# FREAK: A Fine-grained Hallucination Evaluation Benchmark for Advanced MLLMs

**Zhihan Yin**[1]**, Jianxin Liang**[1]**, Yueqian Wang**[1]**, Yifeng Yao**[1]**,
Huishuai Zhang**[1,*]**, Dongyan Zhao** [1,2,*]
[1] Wangxuan Institute of Computer Technology, Peking University
[2] National Engineering Research Center of New Electronic Publishing Technologies
{hansqaq}@stu.pku.edu.cn, {zhanghuishuai, dongyanzhao}@pku.edu.cn

## Abstract

Multimodal Large Language Models (MLLMs) suffer from hallucinations. Existing hallucination evaluation benchmarks are often limited by over-simplified tasks leading to saturated metrics, or insufficient diversity that fails to adequately assess the hallucination extent in state-of-the-art multimodal models. To address this gap, we propose FREAK(**F**ine-g**R**ained **E**valuation **A**gainst **K**nowledge), a comprehensive multimodal benchmark designed for fine-grained hallucination assessment in MLLMs. Through high-quality photorealistic images featuring fine-grained counter-commonsense edits, FREAK innovatively evaluates hallucination phenomena in detailed visual perception of MLLMs. Extensive experiments on FREAK show severe hallucination issues in SOTA models regarding detailed visual perception. To enable deeper investigation, we curate a controlled subset to indirectly evaluate the model's ability to perceive target detailed information. Through systematic evaluation of prevailing Chain-of-Thought (CoT) prompting techniques within this task, we reveal critical insights regarding hallucination patterns and model reasoning processes. The data and code are available at Github.

## 1 Introduction

Multimodal hallucination typically manifests as generated content that, while logically plausible and commonsensical, includes information absent from the visual input and is factually inconsistent with the provided image evidence (Bai et al., 2025; Liu et al., 2024b). Among various forms, one challenging subtype is fine-grained hallucination, where models misperceive or fabricate localized details within an image, often defaulting to commonsense knowledge over visual facts (Wu et al., 2025). Despite significant progress in image comprehension and depth reasoning (OpenAI, 2025; Zheng et al., 2025; Shen et al., 2025), existing MLLMs persistently suffer from multimodal hallucination (Bai et al., 2025), posing a critical gap for stable industrial deployment and everyday use (Magesh et al., 2024). To scientifically quantify the extent of hallucination, prior studies have established dedicated evaluation benchmarks, providing a robust foundation for evaluation.

As the capabilities of MLLMs rapidly advance, their performance on existing hallucination benchmarks such as POPE (Li et al., 2023b) and AMBER (Wang et al., 2024a) has nearly saturated, thereby diminishing the discriminative power of these benchmarks. This saturation largely stems from the limited difficulty of existing benchmarks and their overly simplistic evaluation protocols, which often rely on binary (true/false) judgments. (Tu et al., 2025; Wu et al., 2025), and thus these benchmarks fail to accurately capture the hallucination levels of current SOTA models. As MLLMs are typically trained on large-scale image-text corpora, they are susceptible to data leakage and memorization bias toward specific images. To address this, recent studies have utilized AI-generated **counter-commonsense (CCS) images**, which provide a clear path to test whether a model truly "sees" an image or relies on memorized priors. For example, Guan et al. (2024) created HallusionBench, and Liu et al. (2025) proposed PhD. However, these benchmarks are still limited by insufficient sample diversity, suboptimal image quality, and oversimplified task design.

---

[*]Corresponding Authors

To address the limitations, we introduce **FREAK** (**F**ine-g**R**ained **E**valuation **A**gainst **K**nowledge), which aims to quantify the fine-grained hallucination severity of MLLMs. FREAK consists of 1,799 questions divided into 6 categories, including *Detection*, *Counting*, *Attribute*, *Analysis*, *Position* and *OCR* tasks, providing a comprehensive suite for MLLMs' fine-grained hallucination evaluation. FREAK features its fine-grained CCS questions, high-quality generated images, the diversity of image content, and an objective evaluation methodology.

The construction of FREAK follows a systematic and novel pipeline. First, we instruct LLMs for extensive high-quality CCS description generation. Then, we design a novel "generate-then-edit" process that synthesizes a commonsense-compliant image before using a powerful editing model to introduce a localized, counter-commonsense detail. Next, we leverage LLMs to automatically generate a corresponding question for each image. Finally, this automated data creation is complemented by a crucial human verification and refinement stage. During this step, our team meticulously reviewed each instance and carefully constructed the questions with answers, ensuring free-form questions and multiple-choice questions can directly probe MLLMs' capability for identifying image details and resisting model hallucination.

Extensive experiments on FREAK reveal that even the most advanced models achieve only 45% accuracy, with the performance of mainstream models clustered within the 30%-43% range. This performance falls significantly short of the human baseline (86.71%), highlighting severe fine-grained perceptual hallucination in current MLLMs.

Inspired by advanced reasoning models, we apply CoT prompting across various models but observe consistent performance degradation. Reinforcement learning(RL)-tuned reasoning models also do not show significant improvement over their base versions. By leveraging FREAK, we track model reasoning dynamics and find that **during reasoning process, models exhibit an increasing bias toward distractors and losing confidence in correct answers, often ending with choices contradicting the initial one**. This reveals critical flaws in CoT mechanisms.

In summary, our contributions are as follows.

1. We propose an automated pipeline for generating fine-grained CCS images by integrating LLMs, image generation models and image editing models to produce highly realistic images with localized CCS details.

2. Based on the technical pipeline, we propose FREAK, a novel benchmark to evaluate multimodal fine-grained hallucination. Compared to prior AI-generated hallucination benchmarks, FREAK features an objective evaluation methodology, more diverse CCS descriptions, and more challenging images with questions, revealing critical issues in MLLMs' detail perception capabilities.

3. Extensive experiments on FREAK highlight severe challenges in fine-grained multimodal hallucination for MLLMs. In addition, we discuss the degradation of the CoT prompt, revealing the limitation of CoT reasoning.

## 2 RELATED WORKS

**Multimodal Large Language Models.** Building on rapid advances in LLMs, MLLMs integrating vision and language have also made substantial progress. Current MLLMs achieve visual-linguistic alignment primarily through pretraining or modular training. Some methods develop end-to-end models trained holistically on image-text data (Radford et al., 2021; Li et al., 2021; Cho et al., 2021; Wang et al., 2022). Others preserve frozen LLMs' linguistic abilities while tuning lightweight adapters for cross-modal integration (Liu et al., 2023; Zhu et al., 2023; Li et al., 2023a; Chen et al., 2024d; Bai et al., 2023). This approach avoids costly full-parameter training while leveraging LLMs' generative strengths. For example, BLIP-2 (Li et al., 2023a) uses a Q-Former to bridge visual and textual representations. Competitive alignment can also be achieved via simple linear projectors (Liu et al., 2023; Zhu et al., 2023; Liu et al., 2024c).

**Multimodal Hallucination.** Multimodal hallucination typically manifests as generated content that, while logically plausible and commonsensical, includes information absent from the visual input and is factually inconsistent with the provided image evidence (Bai et al., 2025; Liu et al., 2024b). To mitigate multimodal hallucination, existing approaches fall into two categories: **1)** Designing

Table 1: Comparison between FREAK and other AI-generated benchmarks. FREAK shows uniqueness because of the photorealistic images, fine-grained and diverse counter-commonsense(CCS) content, which strongly challenges SOTA models.

| Benchmark | ImgNum. | Question | GPT Series Eval. | Typical Sample | Explanation |
|---|---|---|---|---|---|
| WHOOPS (Bitton-Guetta et al., 2023) | 500 | VQA | - |  | Einstein uses smart phone. |
| VLind-Bench (il Lee et al., 2025) | 2576 | Y/N | 89.4(GPT-4o) |  | Medieval knight rides motor. |
| PhD-ccs (Liu et al., 2025) | 750 | Y/N | $\sim$79 |  | Max number on dice is seven. |
| HallusionBench (Guan et al., 2024) | 346 | Y/N | 62.28(GPT-4V) |  | Curves have different diameters. |
| VLMBias Vo et al. (2025) | 1392 | Only Count | 20.25(o4-mini) |  | The dog has 5 legs. |
| FREAK | 1786 | Free-Form / MCQ | 42.01(GPT-4.1) |  | The projector is not facing the screen. |

decoding strategies based on heuristic rules to guide models in resisting linguistic priors (Leng et al., 2023; Huang et al., 2024a; Liu et al., 2024d; Wang et al., 2025; Chen et al., 2024c; Zou et al., 2025); **2)** Implementing refined training procedures, such as curating fine-grained image-text data or employing RL-based rules for post-training optimization (Chen et al., 2025; Wu et al., 2024; Yin et al., 2024; Liu et al., 2024a).

**Multimodal Hallucination Benchmark.** Objectively assessing the severity of multimodal hallucination remains a challenging issue. Existing mainstream benchmarks, including POPE (Li et al., 2023b), AMBER (Wang et al., 2024a), MHaluBench (Chen et al., 2024a) and others (Qiu et al., 2024; Wang et al., 2024b; Li et al., 2025) exhibit three critical limitations: **1) Unreliable Data Provenance**: Benchmarks like POPE derive images from open-source datasets that may overlap with training data of evaluated models. Such data contamination risks inflating performance metrics due to model memorization rather than genuine reasoning ability (Chen et al., 2024a; Jiang et al., 2024). **2) Narrow Evaluation Scope**: Traditional large-scale benchmarks predominantly target object hallucination (Chen et al., 2024b; Lovenia et al., 2024), neglecting diverse hallucination types such as OCR, reasoning and object attributes. As a result, SOTA models achieve near-saturation performance, making these benchmarks inadequate for contemporary evaluation. **3) Oversimplified Assessment Paradigm**: Prior evaluations rely heavily on binary true/false judgment (Guan et al., 2024; Huang et al., 2024b), introducing significant randomness. Recent efforts leverage AI-generated CCS images to evaluate the robustness of the models. For example, il Lee et al. (2025); Liu et al. (2025); Guan et al. (2024); Huang et al. (2024b) create such benchmarks with CCS images. However, these benchmarks still suffer from metric saturation, limited hallucination diversity, and synthetic artifacts compromising visual realism (Bitton-Guetta et al., 2023). To address these gaps, we propose FREAK, a hallucination evaluation framework designed for fine-grained hallucination assessment of modern SOTA MLLMs. Unlike recent works like MIRAGE (Dong et al., 2025), LongHalQA (Qiu et al., 2024) that evaluate the long-form outputs of MLLMs, FREAK focuses on specifically targeting MLLMs with fine-grained CCS visual challenges. Table 1 shows the comparison between FREAK and other AI-generated CCS benchmark.

## 3 METHODOLOGY

### 3.1 AUTOMATIC PIPELINE FOR FINE-GRAINED CCS IMAGES

The visual sources for counter-commonsense (CCS) images in prior research mainly stem from two methodologies: **a) Manual Expert Modification** (e.g. HallusionBench (Guan et al., 2024)), which involves human experts altering existing images to introduce contradictions to commonsense knowl-

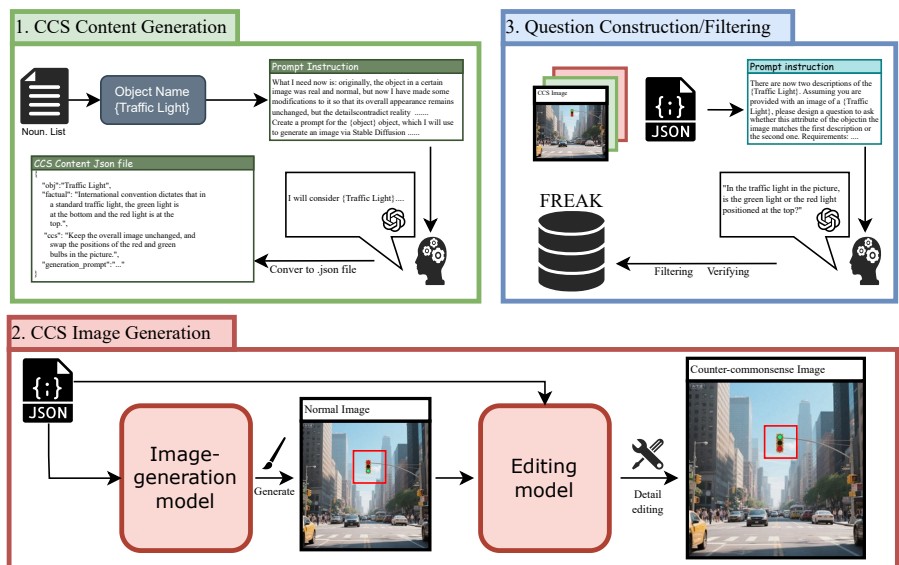

Figure 1: Generation pipeline of FREAK, including a three-step generation paradigm. We first prompt LLMs to generate CCS content for later stages, next we use image genration model and editing model to generate CCS image. Finally we filter and verify the generated data to form FREAK benchmark. In this example, we exhibit the generation process of "Traffic Light". The generated image shows the positions of the red and green lights swapped, incorrectly with red at the bottom and green at the top, which violates both commonsense and the traffic regulations of various countries.

edge; **b) Direct Prompt-to-Image generation** (e.g. PhD (Liu et al., 2025) and WHOOPS (Bitton-Guetta et al., 2023)), where LLMs generate textual descriptions for image generation models. These approaches exhibit critical limitations: Manual modification suffers from low scalability due to its labor-intensive nature, which hinders large-scale dataset construction. On the other hand, directly prompting LLMs to generate CCS descriptions quickly results in repetitive objects and low-quality descriptions, hindering the large-scale production of diverse CCS descriptions. Beyond the description aspect, image generation models show poor adherence to CCS-specific prompts. They predominantly generate commonsense-compliant images due to the lack of CCS-related training data, making it difficult to produce realistic CCS images. In short, image generation models can only reproduce patterns they have already encountered. To address this issue, we deploy the image generation model and editing model in an iterative pipeline: it first generates normal factual images, then applies localized modifications through the editing model for fine-grained CCS details.

### 3.1.1 STEP 1: GENERATION OF CCS DESCRIPTION

To obtain diverse CCS images, it is essential to first generate varied CCS descriptions, such as "a fox with square ears" or "a sofa facing away from a television". For scalable and non-repetitive description creation, we begin by specifying a target object ("fox" and "sofa" in the above examples respectively), and then prompt LLMs to generate attribute descriptions. Finally, we derive a tuple $(O, A, W)$ for subsequent generation stages, where $O$ denotes target object, $A$ denotes correct description of a specific attribute of $O$, $W$ denotes the CCS description of the same attribute.

### 3.1.2 STEP 2: GENERATION OF CCS IMAGES

We employ a two-stage image generation framework. First, we construct a prompt using the target object $O$ and its correct attribute description $A$, then feed it to the image generation model $F$ to produce a normal image: $P = F(O, A)$. Next, we use the image editing model $E$ to modify $P$ conditioned on the CCS description $W$, yielding CCS images: $CCS = E(P, W)$. This framework ensures that the resulting CCS images remain photorealistic while incorporating localized modifications that deliberately contradict commonsense expectations.

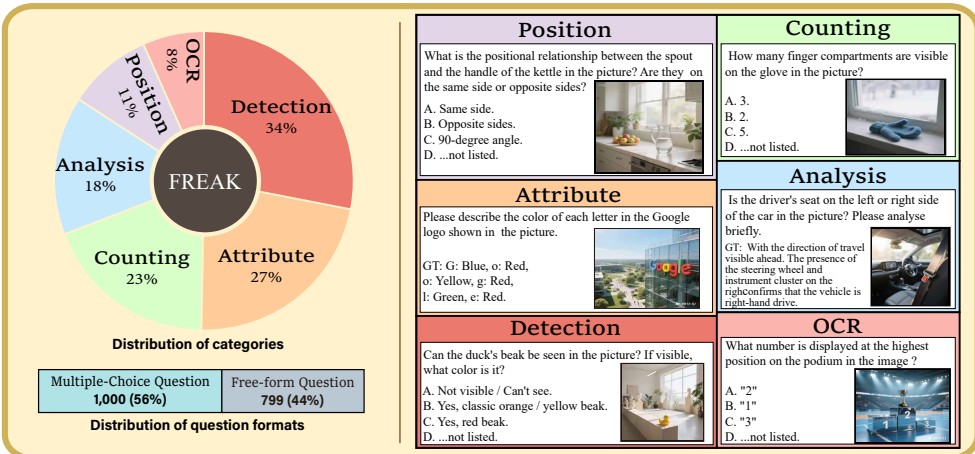

Figure 2: Overview of FREAK. Items in FREAK can be categorized into six subtasks, each comprising questions that are straightforward for human solvers. The right panel shows representative examples for each subtask. Notably, certain questions are assigned to more than one subtask.

### 3.1.3 STEP 3: QUESTION CONSTRUCTION, DATA FILTERING AND HUMAN STUDY

In FREAK, we adopt both **multiple-choice questions** and **free-form questions** as evaluation formats. All questions are generated by LLMs based on $(O, A, W, CCS)$. For free-form questions, the ground truth corresponds to the CCS description $W$, while the hallucinated answer is derived from the commonsense attribute $A$. For multiple-choice questions, each image is paired with one question and four answer options: **A. Correct Option**: Semantically aligned with the counter-commonsense attribute $W$; **B. $A$-based Distractor**: Represents the commonsense attribute $A$ (i.e., the hallucination option); **C. AI-generated Distractor**: Synthetic option derived from semantic prompts of $W$ and $A$; **D. Fixed Open Option**: "The correct answer is not listed". This design measures model robustness against commonsense interference via the $A$-based distractor and the open option. Note that in a minority of cases where multiple plausible commonsense responses may exist, FREAK employs only the dominant distractor for simplified assessment.

After obtaining the tuples $(O, A, W, CCS)$ with questions, we filter the data according to the following rules. **1) Image Filtering**: The selected images must retain photorealistic characteristics, while the image content must be strictly aligned with the CCS description $W$. **2) Deduplication**: We remove duplicate entries with overlapping semantics in $W$ (e.g., "a clock with 4 clock hands" vs. "a clock with 5 clock hands"). This process ensures data diversity and avoids semantic redundancy.

To validate dataset rationality and detect potential biases, we conduct a blind-test experiment with 100 inexperienced undergraduates. We ensured that: (1) participants were unaware of the experiment's purpose, image characteristics, or related content; (2) they were informed that images and questions might be counter-intuitive and were asked to answer faithfully based on the image; (3) each participant randomly answered 17–19 questions to prevent learning effects; and (4) all participants were undergraduate students from various disciplines. These guidelines align with the prompts used for MLLMs to ensure fairness. The experiment establishes a baseline for human performance, quantifying the average performance of untrained humans on the annotated dataset and validates the reliability of the annotated dataset.

## 4 FREAK: FINE-GRAINED EVALUATION AGAINST KNOWLEDGE

For a better understanding of our benchmark, we here analyze its subtasks, statistics and evaluation methodology. The overall composition of FREAK and representative examples for each category are illustrated in Figure 2.

## 4.1 Statistics and Application Tasks

FREAK comprises 1,786 CCS images and 1,799 questions, with 1,000 multiple-choice questions and 799 free-form questions respectively.

To better understand how MLLMs' capabilities change when solving different problems, we divide FREAK into six tasks for cognitive evaluation: **1) Detection**: Requires models to identify salient structures of target objects. In FREAK's CCS images, these structures may be missing or replaced with foreign ones; **2) Counting**: Evaluates models' ability to enumerate the target structures. This task emphasizes hallucination detection rather than numerical proficiency, as more than 90% of the cases contain fewer than six targets; **3) Attribute**: Demands descriptions of geometric attributes (e.g., shape, size, color) for specified structures; **4) Analysis**: Tests the models' inference capabilities based on visual content. The model is required to autonomously locate relevant visual cues after understanding the question. We exclude math and specialized knowledge because they are irrelevant for assessing fine-grained multimodal hallucinations; **5) OCR**: Challenges models to extract target text or identify specified characters. In FREAK, all letters are standard English letters, and we specifically focus on hallucinations in textual content; **6) Position**: Requires the model to determine the spatial locations or relationships of specific objects or structures within the image. We use these tasks to probe MLLMs' abilities through different application-oriented setting. Notably, due to their comprehensive nature, some items in FREAK fall into multiple subtasks. This overlap enables a more objective evaluation of MLLMs' performance across subtasks.

## 4.2 Evaluation

We evaluate models on both free-form and multiple-choice questions. For free-form questions, we adopt the LLM-as-judge approach: an LLM is given with the ground truth and commonsense answer to determine whether the MLLM's response incorporates the CCS content in the image, assigning each response to one of three categories: **Correct**, **Commonsense Error**, or **Other Error**. For multiple-choice questions, correctness is determined directly based on the selected option. The primary performance metric is accuracy ($Acc$), computed as the proportion of correct responses across all questions. To directly measure the influence of the model's parametric knowledge, we define the **Hallucination Rate** ($HalluRate$) as the proportion of cases where the model either outputs a commonsense answer in free-form questions or selects the commonsense distractor in multiple-choice questions.

## 5 Experiment

We conduct experiments to evaluate the effectiveness of FREAK in measuring fine-grained hallucinations of advanced MLLMs. We first describe the experimental setups, then present the main results and key findings. Further in-depth analyses are presented in Section 6.

## 5.1 Experiment Setups

**Model Use**. During the construction of FREAK, we employ Seedream3.0 and SeedEdit3.0 as generation and editing model for its powerful generating and editing capabilities (Gao et al., 2025). We evaluate a diverse set of SOTA models. Proprietary models include Gemini-2.5 series, OpenAI o3, o4-mini, GPT-4.1 and Claude-4.0-Sonnet. Open-source models include Qwen2.5-VL series, InternVL3 series, Kimi-VL-a3b series, GLM-4.5V, Phi-4-multimodal, Skywork-R1V3, MiniCPM-4V, DeepEyes-7B. This selection covers both general-purpose multimodal models and emerging reasoning-specialized architectures, ensuring broad coverage of current capabilities of MLLMs.

**Inference Details**. For multiple-choice questions, we apply **cyclic permutation** across option orders to mitigate randomness and position bias, thereby obtaining more reliable assessments of models' capabilities. We report averaged results over both multiple-choice questions and free-form questions. All models adopt identical prompts during inference.

**Human Baseline**. We recruit 100 undergraduates, each completing only 18 randomly assigned questions in FREAK to prevent experiential bias accumulation during testing. Their aggregated

Table 2: Evaluation results of SOTA MLLMs, which are outperformed by human experts with wide margins. The highest model performance in each column is highlighted in green, and the second-highest is highlighted in blue. The reasoning process is enabled if the MLLM is capable.

| | Accuracy (↑) | | | | | | | HalluRate (↓) | | | | | | |
|---|---|---|---|---|---|---|---|---|---|---|---|---|---|---|
| | Dete. | Count. | Analy. | Attr. | OCR | Pos. | Overall | Dete. | Count. | Analy. | Attr. | OCR | Pos. | Overall |
| **Human Baseline** | 86.93 | 88.65 | 83.44 | 83.92 | 94.24 | 88.08 | 86.71 | 7.19 | 6.76 | 10.94 | 5.22 | 4.32 | 6.22 | 6.95 |
| *Non-Reasoning Models* | | | | | | | | | | | | | | |
| GPT-4.1 | **50.25** | 19.45 | 33.26 | 48.24 | 38.82 | 39.37 | 42.01 | 36.62 | 46.20 | 49.23 | 38.41 | 40.61 | 49.58 | 44.54 |
| InternVL3-78B | 43.03 | 20.12 | 29.65 | 48.49 | 45.15 | 38.41 | 39.32 | 44.11 | 47.75 | 54.91 | 38.74 | 38.59 | 52.65 | 48.76 |
| InternVL3-38B | 40.06 | 17.84 | 28.09 | 46.46 | 46.81 | 37.21 | 37.24 | 44.99 | 48.20 | 57.15 | 40.36 | 33.76 | 46.68 | 48.79 |
| Qwen2.5-VL-72B | 47.23 | 16.84 | 28.09 | 46.58 | 45.70 | 38.22 | 39.39 | 38.43 | 51.60 | 51.77 | 39.36 | 35.29 | 52.76 | 46.82 |
| Qwen2.5-VL-32B | 38.33 | 16.74 | 25.80 | 42.63 | 40.74 | 31.77 | 34.65 | 46.17 | 47.37 | 54.53 | 40.96 | 39.16 | 56.76 | 49.66 |
| Phi-4-multimodal | 39.49 | 18.89 | 25.52 | 36.60 | 37.34 | 32.77 | 33.32 | 38.13 | **34.63** | 51.05 | 37.64 | 31.25 | 47.22 | 42.13 |
| MiniCPM-4V | 46.12 | **24.91** | 30.40 | 48.97 | 41.88 | 40.88 | 41.44 | 36.49 | 40.64 | 48.62 | 31.09 | 37.16 | **37.53** | 41.08 |
| Kimi-VL-A3B-Instruct | 39.69 | 20.98 | 25.69 | 41.01 | 35.65 | 33.76 | 35.04 | 45.25 | 43.17 | 54.59 | 42.27 | 38.55 | 48.51 | 48.49 |
| *Reasoning Models* | | | | | | | | | | | | | | |
| Gemini-2.5-Pro | 47.85 | 23.98 | **35.67** | **56.12** | **56.90** | **43.41** | **45.49** | 37.24 | 41.27 | **46.40** | 29.26 | 24.33 | 45.75 | **40.26** |
| Gemini-2.5-Flash | 44.04 | 20.81 | 32.10 | 48.02 | 48.51 | 35.46 | 40.02 | 40.97 | 44.47 | 48.13 | 36.14 | 30.29 | 52.81 | 44.75 |
| Claude-4.0-Sonnet | 29.93 | 17.48 | 24.96 | 33.95 | 36.20 | 25.45 | 29.85 | 53.22 | 49.49 | 57.56 | 51.30 | 45.17 | 62.73 | 55.64 |
| o3 | 48.96 | 21.14 | 31.54 | 49.89 | 47.30 | 38.77 | 43.00 | 38.34 | 43.59 | 49.64 | 37.36 | 34.76 | 52.50 | 43.67 |
| o4-mini | 44.51 | 21.96 | 30.43 | 47.22 | 46.55 | 35.87 | 40.79 | 40.48 | 42.09 | 50.01 | 38.02 | 36.96 | 55.07 | 44.82 |
| Kimi-VL-A3B-Thinking | 43.77 | 20.22 | 22.86 | 43.94 | 39.42 | 31.82 | 36.82 | 42.94 | 42.51 | 57.16 | 39.57 | 37.83 | 54.77 | 47.31 |
| GLM-4.5V | 47.85 | 19.41 | 26.99 | 47.89 | 56.53 | 37.41 | 41.19 | 39.53 | 47.49 | 55.95 | 37.23 | 27.15 | 52.77 | 46.17 |
| Skywork-R1V3 | 43.67 | 12.31 | 28.61 | 39.57 | 40.87 | 36.88 | 35.50 | 42.04 | 52.78 | 54.70 | 45.82 | 37.45 | 52.70 | 50.28 |
| MiMo-VL-RL2508 | 42.47 | 18.30 | 24.58 | 48.46 | 45.61 | 35.36 | 37.68 | 43.23 | 44.38 | 54.24 | 37.99 | 32.61 | 56.42 | 47.15 |
| DeepEyes | 25.53 | 16.21 | 24.60 | 34.89 | 36.80 | 27.71 | 28.39 | 54.11 | 48.80 | 54.90 | 44.35 | 37.77 | 53.73 | 53.40 |

results establish the human performance baseline. We report detailed inter-annotator agreement statistics (see Section 3). More statistics results about human-blind test can be found in Appendix C.

## 5.2 MAIN RESULTS

Table 2 shows the detailed performance. Based on these results, the key findings are as follows.

**Overall performance gap between humans and MLLMs.** On FREAK, SOTA models achieve only about 45% accuracy, compared to 86% for humans, revealing a gap of roughly 40 percentage points. This indicates that the tasks in FREAK are relatively straightforward for untrained humans, yet remain a major challenge for current MLLMs, reflecting an inconsistency between model intelligence and human reasoning. Furthermore, the HalluRates of most models are close to or even exceed their accuracy across different tasks, highlighting severe weaknesses in fine-grained hallucination control. Figure 3 shows the evaluation results of the full series of Qwen2.5-VL and InternVL3 models. Except for performance degradation at specific sizes, model performance on FREAK generally increases with model size, consistent with the Scaling Law. Interestingly, small models such as MiniCPM-4V and MiMo-VL-RL2508 achieve results comparable to large-scale models, suggesting that reducing hallucination may require an emphasis on model architecture and training processes.

**Uneven performance across tasks.** Breaking results down by task type, models perform worst on *Counting* tasks, while achieving relatively better results on *Attribute* and *OCR* tasks. Although counting appears particularly challenging for MLLMs, most counting questions in FREAK involve small numbers, revealing severe failures in quantity perception. *Attribute* tasks in FREAK primarily comprise shape, color, texture, and other low-level visual tasks. In contrast, *Analysis*, *Position*, and *Detection* questions are predominantly high-level comprehension tasks. Models show better performance on low-level problems, whereas hallucination becomes more severe on high-level tasks. This trend may be explained by the stronger reliance of high-level reasoning on linguistic priors, which causes models to over-rely on their parametric knowledge rather than visual evidence.

Table 3: Comparison of accuracy between Normal Images and CCS Images.

| Model | Size | Normal Img. | CCS Img. |
|-------|------|-------------|----------|
| InternVL3 | 14B | 91.26 | 34.69 (↓56.67) |
| | 38B | 93.63 | 43.97 (↓49.66) |
| Qwen2.5VL | 7B | 86.04 | 34.28 (↓51.76) |
| | 32B | 90.31 | 36.25 (↓54.06) |

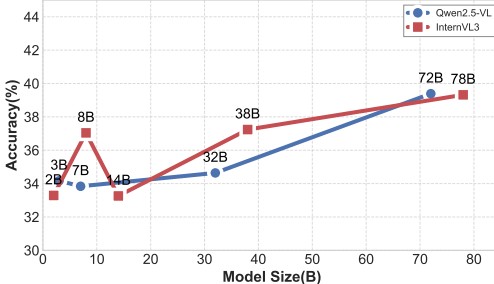

Figure 3: Accuracy evolution across sizes.

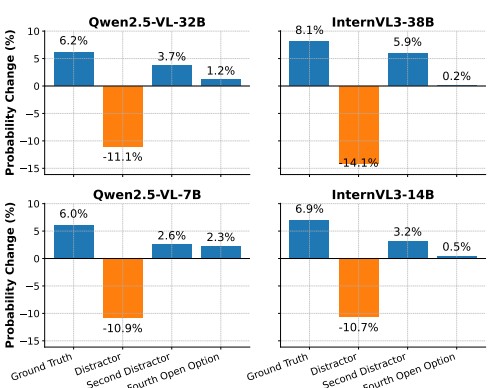

Figure 4: After replacing the standard image with a CCS image, the variation in the model's probability distribution across options for the same error case.The increase in ground-truth probability significantly exceeded that of the other two options, indicating targeted probability enhancement toward correct answers.

Table 4: Performance of different models under two response modes: (1) direct answer generation; (2) reasoning before final output. For non-reasoning models, we employ CoT prompting, while for reasoning models, we activate their reasoning mode. For o3, we use "low" and "high" respectively in reasoning effort parameters of OpenAI API.

| Model | Size | Accuracy (↑) | | HalluRate (↓) | |
|-------|------|--------|-----|--------|-----|
| | | **Direct** | **CoT** | **Direct** | **CoT** |
| GPT-4.1 | - | 42.01 | 40.66(↓1.45) | 45.43 | 46.30 (↑2.65) |
| InternVL3-78B | 78B | 39.32 | 33.91(↓5.41) | 48.76 | 52.83 (↑4.07) |
| InternVL3-38B | 38B | 37.24 | 36.40(↓0.84) | 48.79 | 49.00(↑0.21) |
| Qwen2.5-VL-72B | 72B | 39.39 | 33.39(↓6.00) | 46.82 | 50.95 (↑4.13) |
| Qwen2.5-VL-32B | 32B | 34.65 | 29.82(↓4.83) | 49.66 | 54.52(↑4.86) |
| Phi-4-multimodal | 6B | 33.32 | 25.09(↓8.23) | 42.13 | 46.83(↑4.70) |
| Kimi-VL-A3B-Instruct | 16B | 35.04 | 30.56(↓4.48) | 48.49 | 52.11(↑3.62) |
| Gemini-2.5-Flash | - | 38.10 | 40.02 (↑1.92) | 47.93 | 44.75(↓3.18) |
| o3 | - | 45.15 | 43.00(↓2.15) | 41.53 | 43.67(↑2.14) |
| MiMo-VL-RL2508 | 7B | 41.86 | 37.68(↓4.18) | 43.10 | 47.15(↑4.05) |
| GLM-4.5V | 108B | 41.62 | 41.19(↓0.43) | 46.54 | 46.17(↓0.37) |

**Reasoning process shows no clear advantage.** Reasoning models do not demonstrate significant advantages except for Gemini-2.5-Pro. For instance, among the OpenAI models, o3 improved accuracy by only 1% compared to the non-reasoning model GPT-4.1, while Kimi-VL-A3B-Thinking outperforms the SFT variant by less than 2%. Notably, the small non-reasoning model MiniCPM-4V surpassed all open-source reasoning models. Table 9 shows the performance differences of various models when reasoning before answering versus outputting answers directly. Most models including o3 exhibit varying degrees of metric degradation after activating thinking. The above experimental results indicate that reasoning does not yield a noticeable improvement for multimodal models; on the contrary, most models exhibit performance degradation. We will provide an analysis why reasoning shows no advantage in Section 6

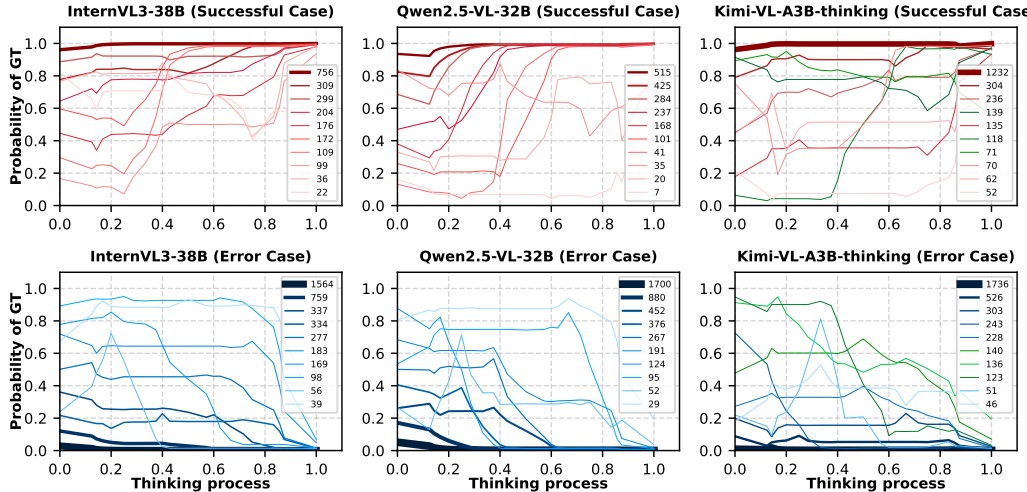

Figure 5: Clustering results showing the evolution of ground-truth probabilities during the reasoning processes. The legend indicates the representative count for each clustered curve. The green curve demonstrates a thought process distinct from conventional MLLMs. Evaluation is conducted on multiple-choice questions with cyclic permutation, where each question is repeated six times.

## 6 ANALYSIS

### 6.1 TEST MLLMS WITH NORMAL / CCS IMAGE PAIR

Can MLLMs truly perceive fine-grained modifications? This is the core question of fine-grained multimodal hallucination. To investigate, we construct a subset of multiple-choice questions where the original data tuple $(O, A, W, CCS)$ is preserved, but the CCS image is replaced with its corresponding normal image $I$. This yields two distinct tuples: $(O, A, W, CCS)$ and $(O, A, W, I)$. We evaluate Qwen2.5-VL and InternVL3 with both types of data, with results summarized in Table 3. Results show that the accuracy drops sharply to around 50% for both models when switching from normal images to CCS images. We further analyze the probability shifts across the four options in error cases. Compared to using normal images, Figure 4 shows that the average selection probability of the distractors decreases by 11% when switching to CCS images. For the remaining three options, the probability distribution exhibits targeted shifts: the correct option receives a substantially larger probability increase than the other two. This may suggest that even in error cases, the models can still extract and comprehend critical information about fine-grained modifications from CCS images. An example of normal and CCS settings can be found in Fig 6.

### 6.2 EXPERIMENT ON CoT REASONING

Table 9 shows that enabling CoT reasoning leads to varying drops in accuracy on FREAK, accompanied by an increase in HalluRate. To further investigate the performance degradation of CoT, we track the evolution of option probabilities during the reasoning process for InternVL3, Qwen2.5-VL, and Kimi-VL-A3B-thinking models. Specifically, we record the probability of the correct option at the end of each reasoning step within the CoT process when the model solves multiple-choice questions. We then apply time-series K-means clustering to group reasoning trajectories across questions to intuitively understand model reasoning patterns. The resulting clusters in Figure 5 reveal two typical failure modes: (1) The model favors an incorrect option from the start and reinforces it throughout reasoning, driving the ground-truth probability close to zero, accounting for **over 70%** of cases; (2) In the remaining cases, the model initially favors the correct answer but abruptly switches to the incorrect one after generating hallucinated content at later stages. Sampling shows that this late-stage hallucination causes correct initial judgments to reverse, finally degrading the performance.

The reasoning trajectories of Kimi-VL-A3B-Thinking, while broadly similar to traditional MLLMs, exhibit more complex patterns (green curves in Figure 5). We suggest that the RL-trained reasoning models enhance their capabilities to perform genuine iterative analysis regarding specific problems through reasoning outputs. However, approximately 77% of Kimi-VL-A3B-Thinking's errors stem from initially incorrect choices whose probability remains unchanged during reasoning. This indicates that only textual reasoning fails to correct text outputs with vision information, resulting in no significant performance gain on FREAK.

Based on the subset analysis and the visualization of reasoning processes across models, we conclude that while MLLMs can perceive the modified CCS information in FREAK, they still tend to rely on internal knowledge and favor distractors. Particularly during the textual reasoning phase, this bias often manifests as late-stage hallucinated content that reinforces incorrect choices. This rigid pattern sharply reduces the probability of selecting correct options and highlights the negative effects of CoT reasoning. We argue that the key to addressing this issue lies in enhancing the model's visual information perception capabilities and adjusting the balance between visual information and MLLMs' parametric knowledge.

## 7 LIMITATION

The primary limitation of FREAK lies in its relatively small scale. Since each item in FREAK is manually verified to ensure quality, further scaling has not yet been achieved. In addition, FREAK relies on external image generation and editing models, which may introduce subtle biases or imperceptible artifacts into the CCS images. The current analysis of CoT under the fine-grained hallucination setting also leaves room for deeper investigation.

As future work, we plan to explore more cost-efficient verification pipelines to scale up the dataset size. To address potential biases introduced by editing models, we provide an ablation study in Appendix D.1.

## 8 CONCLUSION

We propose FREAK, a novel benchmark designed for fine-grained multimodal hallucination evaluation. FREAK features images that violate commonsense only in details, posing significant challenges to current SOTA models and revealing the gap between humans and MLLMs in understanding image details. We further investigated the models' performance on the subset of FREAK and experimentally revealed the limitations of CoT in hallucination evaluation. Like any benchmark, FREAK has limitations such as a relatively small dataset. Nonetheless, FREAK provides new insights for future research and establishes a new standard for hallucination evaluation of MLLMs.

## ACKNOWLEDGMENTS

The work is supported by National Science Foundation of China (No. 62576015, 62576016) and Beijing Natural Science Foundation (L253001).

## ETHICS STATEMENT

This paper proposes a benchmark for fine-grained hallucination evaluation in MLLMs. All data generated during the research are produced by human-aligned LLMs, image generation models and editing models to prevent the biases from human intervention. Note that some data may involve aspects of human culture and commonsense, such as modifying structural details of landmark buildings. To prevent potential discrimination, we have reviewed all data scheduled for public release. The evaluation process of FREAK strives to be transparent and reproducible, adhering to high standards of research integrity and ethical conduct. No personally identifiable information was collected or processed.

## REPRODUCIBILITY STATEMENT

To facilitate reproducibility, we provide all necessary details and materials. Specifically, the dataset generation process and the prompts used are described in Appendix B, while inference setups and experimental implementations are presented in Appendix C. In addition, we include the source code and evaluation outputs of each MLLM in the supplementary materials.

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

## A OVERVIEW OF THE APPENDIX

This appendix is organized as follows: **Section B** discusses about the uniqueness of FREAK and data generation details of FREAK. **Section C** contains experiment details and provides additional experiment results. **Section D.2** contains a ablation study about the used prompt. **Section E** contains additional error cases of different tasks. **Section F** contains the details about the use of LLMs in this paper.

## B FREAK DETAILS

### B.1 UNIQUENESS OF FREAK

FREAK is characterized by its fine-grained CCS content, which poses significant challenges to existing multimodal models. We compare different benchmarks in Table 1. The uniqueness of FREAK lies mainly in the following aspects: **1) Fine-grained editing in realistic images:** As shown in Table 1, images in FREAK appear realistic overall but contain anomalous details. Unlike other benchmarks that often use artistic or illustrative images, images of FREAK are out-of-domain for existing MLLMs, making them particularly challenging. **2) Advanced question design:** Currently, mainstream hallucination benchmarks used in the industry, such as POPE, primarily employ true/false questions as the core evaluation methodology, which involve a high degree of randomness. FREAK uses multiple-choice questions and free-form questions to ensure a flexible and objective evaluation method. **3) Diverse CCS content** FREAK includes six subtasks, with various objects exhibiting different CCS content in different images, enabling a comprehensive evaluation of fine-grained hallucinations in MLLMs. **4) Revealing the gap between humans and models** The questions in FREAK are relatively easy for humans but highly challenging for existing models, highlighting the limitations of current systems while providing an effective benchmark for future academic and industrial research. Unlike other benchmarks that expose model hallucinations through specially designed tasks, FREAK focuses on assessing MLLMs' comprehension of CCS visual information, revealing severe persistent hallucination phenomena. In future work, we will explore more types of CCS modifications, such as temporal CCS phenomena in multi-image sequences.

### B.2 DATA GENERATION DETAILS

The images in FREAK are characterized by their photorealism and localized CCS details, posing significant challenges to model capabilities. The first step involves preparing a noun list to serve as target entities for subsequent CCS image generation. These nouns must correspond to tangible entities. During the construction of FREAK, we utilized the 1,000 labels from ImageNet-1K and prompted LLM to generate objects containing iconic and detailed (i.e., occupying small areas in images) structures, with complex morphological features. By modifying the detailed characteristics of such objects, fine-grained data that contradict commonsense are constructed. While ImageNet-1K includes some fine-grained categories (e.g., Golden Retriever, Labrador, German Shepherd), FREAK is designed to be answerable without requiring domain-specific expertise. It intentionally avoids subtle inter-class distinctions (e.g., that a frilled shark has six gill slits, while a great white shark has five). Therefore, we filtered out overly fine-grained categories, retaining only commonly recognized entities.

For a given entity, we first generate CCS content by using prompts to guide an LLM in producing details related to that entity that contradict common knowledge. The prompt template can be found in Figure 8. Briefly, we instruct the LLM to modify distinctive attributes of the object in a way

that deviates from reality, ensuring the alterations are both adversarial and semantically relevant to the original entity, rather than arbitrary or unrelated. To better control the quality of the generated CCS content, we guide the model to perform edits in several aspects:(1) quantity modification, (2) color and shape alteration, (3) deletion or addition of key structures, (4) replacement of critical components, and (5) logical or physical manipulation that violates real-world constraints or everyday experience. For each type of modification, we provide several examples to help the model correctly understand the desired editing approach and avoid generating "artistic" or surrealistic imaginations. We also require the model to provide step-by-step reasoning during output generation and finally output with JSON format and get tuple $(O, A, W)$, encouraging it to ground its edits in realistic attributes of the object and produce adversarial, high-quality CCS content.

It should be noted that directly instructing models to randomly generate CCS content leads to quickly exhibited repetitive patterns. Furthermore, we observed that the characteristics of CCS content vary significantly across different models.

Subsequently, based on the object $O$ and a correct description $A$ of one of its attributes, we instruct the LLM to generate a prompt for image generation. To ensure scene diversity, the LLM is required to autonomously select appropriate contexts and enrich details of both the scene and the object within it. Since the generated images must be realistic and ordinary, we enforce the inclusion of supplemental terms such as "photorealistic" in the output prompt. The prompt is listed in Figure 9.

Utilizing this image generation prompt, we can synthesize images with normal content. We employed Seedream 3.0 as the image generation and SeedEdit 3.0 as editing model due to its powerful capabilities in generation and modification. We believe that more advanced image generation and editing models can yield better CCS image generation results. Subsequently, we used the generated normal images as input and, combined with the prepared CCS content, performed detailed editing on the original images. Limited by the current capabilities of image generation models, even when using image editing models to introduce CCS modifications rather than directly generating CCS images, the resulting pictures may still fail to incorporate the required CCS content. Therefore, after obtaining CCS images, manual screening is necessary. The screening criteria primarily focus on three aspects: (1) The CCS content must be valid and reasonable. Since the quality of LLM-generated CCS content varies, we require that the CCS aspects in the images must correspond to detailed visual features. (2) The CCS modifications should not be highly repetitive; for example, not all animals should have their leg counts altered. (3) The images must be realistic and the errors clearly identifiable, avoiding ambiguity or misleading appearances. Finally, the screened images, along with the originally generated content from the LLM, are used to instruct the LLM to generate distracting options, thereby forming a complete instance.

To ensure alignment with human preferences and mitigate potential biases introduced during the annotation process, we employed a human baseline approach to evaluate the validity of the data. We recruited 100 ordinary university students, each assigned only 18 randomly selected questions to prevent experience accumulation during the task. Without prior contextual hints, these participants achieved an accuracy rate of 86.71%, which empirically validates that the benchmark data aligns well with human reasoning preferences and effectively avoids potential biases.

Our generation pipeline demonstrates high scalability: given only object names, it synthesizes diverse CCS images for each target objects. During FREAK's construction, we directly leverage the ImageNet-1K label set as the object source, and prompt LLMs to generate approximately 1,500 entities to compensate for objects absent in ImageNet such as landmarks, famous branded products, and various foods. By substituting other noun collections (e.g., domain-specific lexicons or larger label list), the pipeline achieves zero-shot generalization to new object categories for CCS image generation. This implies that FREAK's data generation approach can be extended to a wider variety of entity types, ensuring scalability of the data. Moreover, this method allows room for further design of more fine-grained hallucinations. For instance, by creating subtle morphological differences among different species within the same biological category to generate more challenging data samples.

Table 5: Human performance metrics on the multiple-choice task. We randomly shuffled the order of the answer choices and ensured that the number of questions with A, B, and C as the correct answer was balanced. Option D was fixed as: 'Correct answer is not listed.'

| Choice | Precision | Recall | F1-score | Support |
|---|---|---|---|---|
| Choice A | 0.83 | 0.86 | 0.84 | 340 |
| Choice B | 0.86 | 0.86 | 0.86 | 335 |
| Choice C | 0.87 | 0.83 | 0.85 | 335 |
| Accuracy | 0.85 | 0.85 | 0.85 | 1010 |
| Macro Avg | 0.64 | 0.64 | 0.64 | 1010 |
| Weighted Avg | 0.86 | 0.85 | 0.85 | 1010 |

## C    EXPERIMENT DETAILS

In this section, we will detail the implementation, especially the implementation details of experiments in Section 6.

### C.1    MAIN EXPERIMENT DETAILS

For open-source models, we use vLLM-0.10.1 for inference, for closed-source models, we use official API. When making requests to the model, we consistently use the parameters temperature=0 and seed=42 to ensure reproducibility. For the main experiment, we use the prompt in Figure 10. For CoT reasoning, we use the prompt in Figure 11

#### C.1.1    HUMAN BLIND TEST DETAILS

We conducted a human blind test to validate the reliability of our dataset and to establish a human performance baseline for model comparison. To ensure a fair and controlled experimental setup, we enforced the following conditions: (1) participants were not informed of the experiment's purpose, the characteristics of the images, or any details related to the questions; (2) participants were explicitly told that some images and questions might be counter-intuitive and were instructed to answer strictly based on the visual content; (3) each participant answered only 17–19 randomly selected questions to avoid familiarity effects; (4) all participants were undergraduate students from diverse academic majors; (5) each question was answered by one or two participants, ensuring full coverage of all items. Given the number of participants and by the central limit theorem, the aggregated responses can be considered representative of human choices for these tasks.

These instructions were designed to align with the prompting conditions used for MLLMs and thus avoid any unfairness. We report the human results in the main table, and for transparency and reproducibility, we provide additional metrics for the multiple-choice questions in Table 5.

#### C.1.2    MORE EXPERIMENT RESULTS OF MULTIPLE-CHOICE QUESTIONS

We use Cyclic Permutation for evaluation of multiple-choice question. We repeated each question six times, each time altering the permutation order of the options. Since we have three options besides option D, we generated all possible permutations of these three options, resulting in six repetitions for each original question. Note that for each repetition, the labels preceding the options strictly adhere to the sequence A, B, C, D, with only the content of the options being swapped. For model outputs, we use regular expressions to extract the model's selections. For instances where certain models deviate from the specified output format, given that FREAK aims to evaluate the degree of hallucination rather than instruction-following capability, we employ GPT-4o-mini to assist in judging and selecting the correct answer based on the unformatted content, thereby avoiding mis-

Table 6: More evaluation results on FREAK, including **Accuracy**, **Consistency Accuracy**, **Weighted-Precision**, **Weighted-Recall**, and **Weighted-F1**. It should be noted that due to the use of cyclic permutation, the support for options A, B, and C is 2000 each, whereas the support for option D is 0. After excluding option D, the weighted metrics are equivalent to the macro-averaged metrics.

| Model | Size | Accuracy | Consist. Acc. | Precision | Recall | F1 |
|---|---|---|---|---|---|---|
| o3 | - | 43.98 | 32.40 | 45.08 | 43.98 | 44.52 |
| Gemini2.5 flash | - | 41.00 | 25.10 | 42.84 | 41.00 | 41.88 |
| Gemini2.5 pro | - | 42.73 | 33.00 | 45.24 | 42.73 | 43.95 |
| o4-mini | - | 41.58 | 27.60 | 42.92 | 41.58 | 42.24 |
| GPT 4.1 | - | 43.78 | 31.50 | 45.51 | 43.78 | 44.45 |
| Claude4 sonnet | - | 30.30 | 18.50 | 30.60 | 30.30 | 30.42 |
| InternVL3-78B | 78B | 43.63 | **35.90** | 43.65 | 43.63 | 43.63 |
| InternVL3-38B | 38B | 42.60 | 35.00 | 42.69 | 42.60 | 42.62 |
| InternVL3-14B | 14B | 36.03 | 25.30 | 36.10 | 36.03 | 36.01 |
| InternVL3-8B | 8B | 41.23 | 24.80 | 41.64 | 41.23 | 40.86 |
| InternVL3-2B | 2B | 39.58 | 26.20 | 41.86 | 39.58 | 40.61 |
| Qwen2.5-VL-72B | 72B | 36.97 | 30.40 | 37.82 | 36.97 | 37.37 |
| Qwen2.5-VL-32B | 32B | 36.03 | 28.30 | 36.23 | 36.03 | 36.13 |
| Qwen2.5-VL-7B | 7B | 36.08 | 26.00 | 36.67 | 36.08 | 36.34 |
| Qwen2.5-VL-3B | 3B | 36.20 | 25.90 | 36.65 | 36.20 | 36.39 |
| Phi 4 multimodal | 6B | 37.56 | 19.20 | 39.10 | 37.56 | 38.07 |
| Kimi-VL-A3B-Instruct | 16B | 39.23 | 24.70 | 39.29 | 39.23 | 39.20 |
| MiniCPM 4V | 4B | **46.06** | 34.50 | **46.29** | **46.05** | **46.13** |
| Kimi-VL-A3B-Thin | 16B | 40.23 | 24.60 | 41.32 | 40.23 | 40.49 |
| MiMo-VL-RL | 7B | 42.50 | 29.80 | 42.79 | 42.50 | 42.42 |
| GLM 4.5V | 108B | 42.78 | 34.30 | 42.84 | 42.78 | 42.79 |
| Skywork R1V3 | 38B | 36.57 | 22.20 | 37.07 | 36.57 | 36.82 |
| DeepEyes | 7B | 28.67 | 18.60 | 28.78 | 28.67 | 28.72 |

judgments caused by formatting errors. The prompt used for this auxiliary evaluation is provided in Figure 15.

Moreover, the precision, recall and F1 score of multiple-choice questions are measured in Table 6 for reference. From the results in the table, it can be observed that compared to the Average Accuracy, the Consistency Accuracy metric shows a significant decrease across all models, indicating that the models are considerably affected by the permutation of answer options and exhibit notable positional non-robustness. This indirectly reflects the challenging nature of FREAK for the models, as well as their low certainty in answering the questions. The InternVL3-78B model achieved the highest Consistency Accuracy of 35.90, and within the same model series, Consistency Accuracy increases with model size, demonstrating a clear positive correlation and reflecting the effectiveness of Scaling Laws. In contrast, smaller parameter models experienced a more substantial decline in Consistency Accuracy, such as MiniCPM-V4 and Phi-4 Multimodal, implying that limited parameter size leads to weaker stability and robustness.

For Precision, Recall, and F1 metrics, we additionally present the Precision, Recall, and F1 scores corresponding to options A, B, and C in Table 7. It is evident that some models exhibit significant differences in Precision and Recall across different options. Despite the use of Cyclic Permutation, model performance still varies under different option orders. This issue persists even in state-of-the-art models like GPT-4.1.

### C.1.3 LLM-AS-JUDGE DETAILS

We use LLMs to evaluate MLLMs' performance on free-form question. Specifically, we employ GPT-5-mini as the judge model for the evaluation of FREAK, as the tasks in FREAK do not involve complex reasoning or computations. We instructed GPT-5-mini to categorize the outputs of VLMs into exactly three classes based on the provided image, question, correct answer, and commonsense answer: **1) Correct:** The model's output aligns with the image and the corresponding question. **2) Commonsense Error:** The model produced a commonsense answer, which in the context of FREAK contradicts the correct answer. **3) Other Error:** The model generated other types of incorrect content, which often occur during counting tasks. This approach does not require the

Table 7: The Precision, Recall, and F1 scores evaluation results of various models across the three categories of options A, B, and C. Some models exhibit noticeable variations in performance across different options, indicating the presence of certain option order biases under our task design.

| Model | Precision | | | Recall | | | F1 | | |
|---|---|---|---|---|---|---|---|---|---|
| | Op. A | Op. B | Op. C | Op. A | Op. B | Op. C | Op. A | Op. B | Op. C |
| o3 | 44.47 | 45.28 | 45.49 | 43.20 | 44.35 | 44.40 | 43.82 | 44.81 | 44.94 |
| o4-mini | 42.58 | 43.35 | 42.82 | 41.45 | 41.70 | 41.60 | 42.01 | 42.51 | 42.20 |
| GPT 4.1 | 44.69 | 46.15 | 45.68 | 50.95 | 40.45 | 39.95 | 47.62 | 43.11 | 42.62 |
| Gemini2.5 pro | 44.46 | 45.85 | 45.42 | 42.50 | 42.55 | 43.15 | 43.46 | 44.14 | 44.26 |
| Gemini2.5 flash | 42.96 | 43.19 | 42.36 | 43.50 | 39.00 | 40.50 | 43.23 | 40.99 | 41.41 |
| Claude4 sonnet | 29.98 | 30.66 | 31.15 | 27.10 | 31.20 | 32.60 | 28.47 | 30.93 | 31.86 |
| InternVL3-78B | 44.34 | 43.88 | 42.74 | 43.30 | 43.00 | 44.60 | 43.81 | 43.43 | 43.65 |
| InternVL3-38B | 42.96 | 42.69 | 42.43 | 40.00 | 44.10 | 43.70 | 41.43 | 43.38 | 43.05 |
| InternVL3-14B | 35.89 | 36.27 | 36.13 | 32.25 | 39.30 | 36.55 | 33.97 | 37.72 | 36.34 |
| InternVL3-8B | 39.47 | 43.38 | 42.08 | 49.55 | 30.45 | 43.70 | 43.94 | 35.78 | 42.87 |
| InternVL3-2B | 42.20 | 40.98 | 42.41 | 38.40 | 43.50 | 36.85 | 40.21 | 42.20 | 39.43 |
| Qwen2.5-VL-72B | 37.95 | 38.04 | 37.47 | 35.10 | 37.15 | 38.65 | 36.47 | 37.59 | 38.05 |
| Qwen2.5-VL-32B | 36.31 | 36.24 | 36.13 | 36.80 | 34.95 | 36.35 | 36.55 | 35.58 | 36.24 |
| Qwen2.5-VL-7B | 36.74 | 36.36 | 36.91 | 35.25 | 38.40 | 34.60 | 35.98 | 37.35 | 35.72 |
| Qwen2.5-VL-3B | 36.48 | 36.56 | 36.90 | 34.05 | 39.05 | 35.50 | 35.22 | 37.77 | 36.19 |
| Phi 4V | 37.87 | 38.27 | 41.16 | 39.90 | 41.45 | 31.30 | 38.86 | 39.80 | 35.56 |
| MiniCPM 4V | 46.52 | 46.72 | 45.62 | 43.50 | 45.90 | 48.75 | 44.96 | 46.31 | 47.14 |
| Kimi VL A3B(instruct) | 39.23 | 39.11 | 39.54 | 43.00 | 38.80 | 35.90 | 41.03 | 38.96 | 37.63 |
| Kimi VL A3B(thinking) | 40.76 | 41.51 | 41.68 | 49.20 | 37.05 | 34.45 | 44.59 | 39.15 | 37.72 |
| GLM 4.5V | 43.27 | 42.26 | 42.98 | 41.65 | 44.90 | 41.80 | 42.45 | 43.54 | 42.38 |
| Skywork R1V3 | 37.08 | 36.56 | 37.58 | 36.45 | 36.25 | 37.00 | 36.76 | 36.40 | 37.29 |
| MiMo-VL-RL2508 | 41.72 | 42.93 | 43.73 | 49.35 | 42.05 | 36.10 | 45.21 | 42.49 | 39.55 |
| DeepEyes | 28.40 | 29.02 | 28.92 | 28.00 | 29.15 | 28.85 | 28.20 | 29.08 | 28.89 |

LLM to output complex scores, aiming to maintain the objectivity of the LLM evaluation through a simplified method.

To further investigate the consistency between LLM-as-a-judge and human assessments, we randomly sample 100-110 questions for tested models and compared the human evaluation results with the LLM evaluation results. Table 8 shows the calculated consistency between humans and the LLM across different models. From the table, the consistency rate between LLMs and humans exceeds 90% across different models, including both open-source and closed-source models, general MLLMs, and reasoning models, demonstrating a relatively strong alignment. Additionally, the standard deviation of consistency rates is relatively small, with confidence intervals distributed at the high end, further indicating the reliability of LLM-as-judge in the FREAK evaluation. This demonstrates the effectiveness and rationality of using LLM-as-judge for evaluating free-form questions in the FREAK framework.

It should be noted that GPT-5-mini's understanding of the images in FREAK is also not entirely accurate. Although we provided the images to the LLMs during the evaluation process, we instructed the models to derive only a coarse-grained understanding from the images. The LLMs were strictly required to make judgments based solely on the provided text.

## C.2 SUBSET EXPERIMENT DETAILS

In Section 6, we collect a subset that contains both normal images and CCS images. The subset has contains pieces of data. We conduct a controlled experiment on the subset. Figure 6 shows an example of the subset.

We first query the model using normal images, then repeat the same questions with CCS Images to compare changes in the model's responses. As shown in Section 6, for cases where the model errs after switching to CCS Images, we analyze the probability shifts among the four options,which

Table 8: The statistical results of sampling for consistency between the LLM and human evaluators. **Consistency Rate** refers to the proportion of instances where the LLM's evaluation results align with those of human assessors. The **P/N Consistency Rate** consolidates **Commonsense Error** and **Other Error** into a single category, considering only two types of judgments before calculating the consistency proportion. We use GPT-5-mini(2025-08-07) as the judge model in this study.

| Model | Consistency Rate | P/N Consistency Rate | Sample Size |
|---|---|---|---|
| Gemini2.5 pro | 87.74 | 87.74 | 106 |
| Gemini2.5 flash | 91.00 | 95.00 | 100 |
| o3 | 86.54 | 94.23 | 104 |
| GPT 4.1 | 94.12 | 95.10 | 102 |
| InternVL3-78B | 88.24 | 92.16 | 102 |
| InternVL3-38B | 92.00 | 96.00 | 100 |
| Qwen2.5-VL-72B | 95.05 | 95.05 | 101 |
| Qwen2.5-VL-32B | 90.00 | 94.00 | 100 |
| Phi 4-multimodal | 89.11 | 98.02 | 101 |
| MiniCPM-V4 | 95.00 | 97.00 | 100 |
| MiMo-VL-RL2508 | 93.00 | 96.00 | 100 |
| GLM 4.5V | 92.11 | 99.12 | 114 |
| Kimi-VL-A3B-Thinking | 90.10 | 95.05 | 101 |
| **Mean** | 91.08 | 94.96 | |
| **Std. dev.** | 2.66 | 2.70 | |
| 95% **CI** | (89.40, 92.75) | (93.26, 96.65) | |

reveals a directional pattern: even when the model still selects distractor, the probability of the correct option increases, and does so more significantly than other options. This suggests that the model can perceive CCS clues in CCS Images, yet structural or inherent limitations still lead it to choose incorrect options, reflecting severe hallucination. Fig. 7 illustrates the proportional changes of different sample types before and after image substitution: **TP**: Correct before (with Normal Image) and after substitution; **TN**: Correct before, but incorrect after; **FP**: Incorrect before, but correct after; **FN**: Incorrect both before and after. Combined with the results of Table 3, only $\frac{1}{3}$ to $\frac{1}{2}$ of the cases receive the correct responses from the model after switching to the CCS images, while $\frac{1}{2}$ to $\frac{2}{3}$ of the cases remain incorrect.

## C.3 PROBE EXPERIMENT DETAILS

In Section 6, we analyze the probability of ground truth in multiple-choice questions during the reasoning process. To track changes in model preferences during the reasoning process, we first modified the input prompt by removing the few-shot examples (as shown in Figure 12) to avoid constraining or interfering with the model's thinking patterns. By probing the output probabilities of each option during the reasoning process, we can directly analyze the causes of performance degradation in CoT reasoning. This is a key advantage of multiple-choice or closed-ended questions.

We begin by using the prompt to guide the model to output both the entire reasoning process and the final answer. The reasoning process is then split at the sentence level and reassembled using a "prefix-sum" style algorithm, forming a cumulative sequence of reasoning steps. At the end of each intermediate step, we append the phrase "So I will choose <answer>" to simulate the model's concluded thought. This approach mimics the model's own output and allows us to capture its evolving preference at various intermediate stages. Crucially, it ensures that the subsequent output strictly corresponds to the model's choice rather than other content.

In practice, this method is model-agnostic: the original input prompt (without simulated model output) is first wrapped in its chat template, and the simulated output is appended directly to the prompt wrapped in the chat template. Note that simply making the simulated output as the 'assistant'

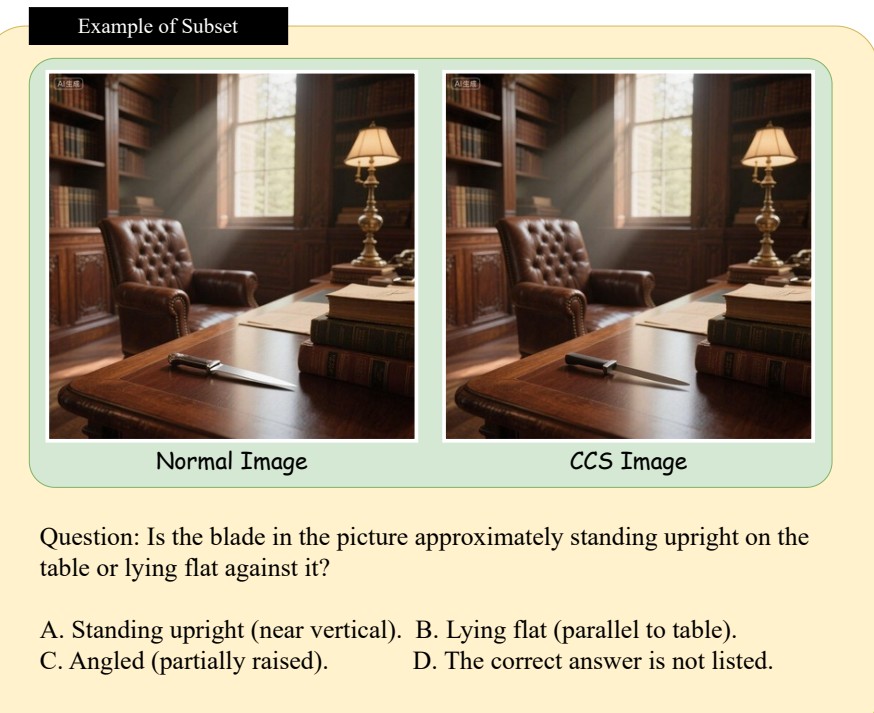

Figure 6: Prompt used for LLM-assisted option selection. For models that generated reasoning but failed to output the final choice in the required format, we used advanced LLMs to make the selection based on their reasoning traces.

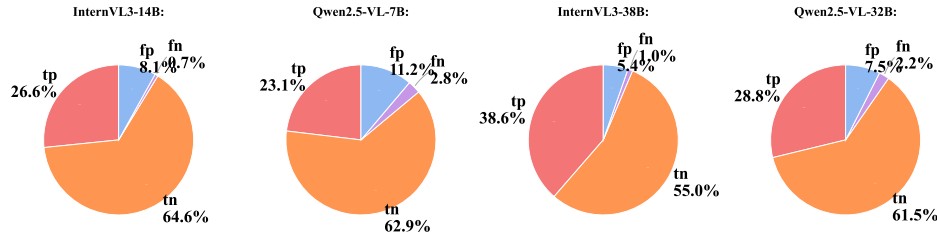

Figure 7: Proportion of different type of instances in subset. **TP**: Correct before (with normal image) and after substitution; **TN**: Correct before, but incorrect after; **FP**: Incorrect before, but correct after; **FN**: Incorrect both before and after.

role prompt is ineffective, which will be regarded as an entire sentence in fact, and model will not continue to generate output based on the simulated content.

This approach is equally applicable to reasoning models. By modifying the appended content to include a termination token for reasoning, we can simulate the model's self-termination of the reasoning process. For example, in the Kimi-VL-A3B-Thinking model, the tokens ◁think▷ and ◁ /think▷ are used to demarcate the thinking process.

Figure 25 shows the full result of the experiment, the analysis is similar to the part in Section 6.

Table 9: Ablation study on editing models. We use different image editing models and evaluate three MLLMs with CCS images generated by these editing models. Comprehensive results indicates that using of SeedEdit 3.0 doesn't introduce structure bias.

| Model | SeedEdit 3.0 | | Nano Banana | | Seedream 4.0 | |
|---|---|---|---|---|---|---|
| | Accuracy | Consitency | Accuracy | Consitency | Accuracy | Consitency |
| Gemini-2.5-flash | 46.23 | - | 44.78 | 83.18 | 47.54 | 79.99 |
| o4-mini | 45.86 | - | 44.20 | 83.04 | 50.14 | 83.48 |
| Qwen2.5-VL-32B | 38.26 | - | 38.12 | 82.32 | 39.71 | 86.96 |

# D    ABLATION STUDY

## D.1    EDITING MODEL ABLATION STUDY

To assess whether the benchmark quality is overly dependent on SeedEdit 3.0, we conducted an ablation study using three different image-editing models: SeedEdit 3.0, Nano Banana, and Seedream 4.0. We randomly sampled 114 MCQ items from FREAK, applied the same editing prompt to all three models, and evaluated the resulting images using three representative MLLMs (Gemini-2.5-flash, o4-mini, and Qwen2.5-VL-32B). As shown in Table 9, the answer consistency between SeedEdit 3.0 and the other editors ranges from 80% to 87%, while the accuracy differences for each MLLM remain small. These findings suggest that although different editors may vary slightly in CCS editing, the downstream difficulty of images and questions remains largely stable. Therefore, benchmark performance is not dominated or distorted by the specific choice of SeedEdit 3.0, and potential information leakage or corruption unique to this editor can be ruled out.

Importantly, during dataset construction, all edited images underwent manual quality verification, ensuring that only visually natural and semantically coherent images were included in the benchmark. This further mitigates concerns regarding artifacts introduced by any single editing model.

For the consistency rate, We analyse that even under the same textual modification prompt, different editing models, and sometimes even the same model across trials, may produce images that are semantically aligned but visually implemented in different ways. Since text prompts typically contain less information than the visual space they condition, the generated images can vary in compositional details despite fulfilling the same modification instruction. These subtle variations alter the difficulty of the corresponding QA, which results in shifts in the MLLMs' performance across editing models.

## D.2    PROMPT ABLATION STUDY

During the evaluation process, we employ two types of prompts: one requires the model to directly output the selected answer to the question, and another requires the model to reason about the question before generating the answer. All models consistently used the same prompt content. For certain models with specific prompt formatting requirements, we only adjusted the format while keeping the content essentially unchanged.

To eliminate the potential impact of our self-designed CoT prompt, we additionally employ the CoT prompt from Figure 13, which has been validated by previous work to enhance model's performance. The evaluation results of each model on FREAK's multiple-choice question seta are presented in Table 10.

The data in the table show that the model's accuracy decreased to varying degrees under the two CoT prompts. Combined with the results from Table 9, conventional MLLMs were more adversely affected by CoT prompting. We have analyzed the reasons for this performance degradation with CoT prompts in Section 6.

Table 10: Result of CoT prompt ablation experiment. **Acc(D)**: Accuracy using direct prompt (Fig. 10); **Acc(v1)**: Accuracy using CoT prompt v1 (Fig. 11); **Acc(v2)**: Accuracy using CoT prompt v2 (Fig. 13).

| Model | Size | Acc(D) | Acc(v1) | Acc(v2) |
|---|---|---|---|---|
| InternVL3-38B | 38B | 42.60 | 40.28 | 36.74 |
| InternVL3-14B | 14B | 36.03 | 35.13 | 28.42 |
| Qwen2.5-VL-32B | 32B | 36.03 | 29.95 | 29.50 |
| Qwen2.5-VL-7B | 7B | 36.08 | 32.98 | 33.32 |
| Phi4 multimodal | 6B | 37.55 | 29.13 | 25.08 |

# E  ERROR CASES

We now present additional case studies showcasing erroneous responses from InternVL3-38B (Figures 16 to 21), and the reasoning model Kimi-VL-A3B-Thinking (Figures 22 to 24). We provide the original questions, corresponding images, options, correct answers, and the models' incorrect outputs. These cases include the models' reasoning processes, revealing that errors in describing target objects in the images led to incorrect choices. For Kimi-VL-A3B-Thinking, a self-reflective reasoning pattern emerged: the model engaged in repeated deliberation, exhibited hesitation during reasoning, and even negated its own intermediate conclusions. We posit that this pattern arises because the model perceives CCS details in the image, which conflict with its internal knowledge acquired during training, thereby inducing self-doubt.

The self-reflect mode observed in the Kimi model provides further evidence of its ability to perceive CCS visual details: when such visual information conflicts with its parametric knowledge, the model exhibits self-reflective reasoning. However, after deliberating for a certain length, it still generates hallucinated content and ultimately selects incorrect answers. This suggests that mitigating hallucinations requires enhancing the model's understanding and trust in the visual information it captures, preventing internal knowledge from overriding genuine visual cues. We argue that the key lies in improving the alignment between the visual encoder and the LLM. Simply enhancing reasoning capacity without strengthening visual understanding, as seen in Kimi-VL-A3B-Thinking, allows the model to detect inconsistencies but not to arrive at correct answers.

# F  THE USE OF LLMS

We employ LLMs in our data generation pipeline for FREAK to produce high-quality CCS content, for which we have provided the relevant prompts. Beyond this specific application, LLMs are used only for reference in the writing of individual words and sentences in the paper. We further assure that the methodological ideas are independently conceived by the authors, and all experiments are conducted independently without the use of LLMs for research assistance.

---

**Prompt: CCS Content Generation**

The content I require is as follows: An object in an original image is realistic and normal, but I have modified it in a way that preserves its overall structure while introducing detail-levelinconsistencies with reality, making it implausible. Below, I will first introduce several common modification methods.
#######################################
1. Quantity Modification: Add or remove a key structural element of the object so that the result contradicts reality. Note: Modifications must be based on the distinctive features of the specific object, not applied generically (e.g., not just arbitrarily changing the number of legs on an animal).

Example: ① For a starfish: Add an arm, creating a six-armed starfish. (Explanation: Real-world starfish have five arms; this modification is subtle regarding the image and object.) ② For a western dining fork: Change the fork from four tines to five. (Explanation: Standard western forks have four tines; five is implausible and a detail-level change.) ③ For a clock: Change the clock's hands to four pointers. (Explanation: Clocks with four pointers do not exist in reality.) ④ For a snowflake: Change the snowflake from six branches to eight. (Explanation: Due to water molecule structure, real snowflakes always have six branches; this modification focuses on branch count detail.) ⑤ For a guitar: Change the guitar from six strings to five. (Explanation: Standard guitars have six strings; changing the string count is a detail modification.)
#######################################
2. Color Modification: Alter the color of a small partof the object, creating a detail that contradicts the real world. Note: The color change must focus on a very small area; large-area changes (e.g., turning all a zebra's stripes black) are not allowed, as this makes it a different ordinary object (a horse) and violates the requirement that the modified object remains implausible. Note: The color change must result in an object that defies common sense.

Example: ① For a rainbow: Change the innermost color to red. (Explanation: Real rainbows strictly follow the order red-orange-yellow-green-blue-indigo-violet from the outside in; the innermost cannot be red, and this change affects only one layer.) ② For a traffic light: Change the bottom bulb to blue. (Explanation: Traffic lights are red, yellow, green from top to bottom; changing the bottom green to blue contradicts reality.)
(For simplicity, We omit other categories)

The above only lists six types, but other modifications violating common sense exist (e.g., a playing card printed with several suits, a compass with nine directional markers). You need to think carefully and cleverly based on the object's characteristics to devise suitable modifications.

Here, the object you need to consider is: {object}. You should first analyze its characteristics, including color, shape, key structures, and their quantities. Indicate which one or severalmodification strategies are possible. After careful and meticulous thought, list the feasible modification methods for this object. Your response must follow this format:

[object]: [Describe the object in one sentence, including the various features mentioned above]. Analysis: [In-depth analysis of which aspects can be modified].
Format: <answer><condition>[The required state of the object in the image, determined by the modification rule. E.g., to delete a piano's black keys, write: Overhead view of a piano, black keys clearly visible.]</condition> <rule>[Modification rule: delete, add, or modify a specific feature of the object in the image]</rule><description>[A reasonable explanation for this modification] </description><name>[Filename to save]</name></answer>

Requirements:

1.
Modifications cannot be too difficult or far-fetched. They must allow a normal person to notice the issue in the modified image. Provide only 2-4 of the best, most clever modifications (refer to the examples). If the current object has no worthwhile subtle modifications, skip it directly!

2.
Your modifications cannot be too specialized or academic!! For a specific object (creature, tool), you should choose to modify its characteristics as a member of a broader category. For example, for a tiger shark, modify its characteristics as a shark, or even as a fish(e.g., number of dorsal/ventral fins)! Do notmodify features specific to its very niche category, as this will make it impossible for humans to judge! The criterion is: Can an average person (without professional taxonomic training) recognize this anomaly? In the previous example, oridinary people cannot distinguish tiger sharks, so you cannot modify features specific to tiger sharks (e.g., tiger sharks have five gill slits).

....(For simplicity, we omit other rules)...

Now, please analyze the {object}object according to the guidance above and the examples. Suggest what modifications are possible. Respond strictly in the required format. Do not add any extra text.

---

Figure 8: Prompt used for CCS content generation

---

**Prompt: CCS Content Generation**

Create a prompt for the {object} object. I will use this prompt to generate an image via Stable Diffusion.

Note: Do not deliberately emphasize the characteristics of the {object}. The generated scene must meet the requirements in parentheses while ensuring a photorealistic effect. Additionally: Do not use depth-of-field-related terms in the prompt. Also, require that the {object} is at a certain distance from the cameraand occupies only a small part of the image(you must explicitly state this in the prompt!). Determine this distance appropriately based on the object. Furthermore, you need to specify a suitable scene for this object, where the object is only one part of the scene.

Note: Generate only the prompt itself, without any additional explanation. If this is a logo, generate the prompt for the corresponding product/object!

Example: Hyper-realistic acoustic guitar (six strings clearly visible), positioned on the left side of the image, tilted on a wooden desk in a sunlit study, warm ambient lighting, placed at a distance from the camera, with a small speaker next to the guitar.

---

Figure 9: Prompt used for image generation.

---

**Prompt:directly require choice (main experiment)**

system:
"You are a helpful agent.Here is an image with a multiple choice question about the image content. You should reply the question according to the image faithfully. Please note that the question maybe confusing or the image content might be uncommon, you should answer the question ONLY with the correct choice letter.
Here is an example:
#########
[IMAGE]
Question:Does the Teapot in the picture have a handle? If so, where is it located?
Choices:
A. Not visible / Can't see.
B. Yes, on the side.
C. Yes, arched over the top.
D. The correct answer is not listed.

Your answer: A
#########
Now please answer the question following the above format STRICTLY."

user:
<image>
"Question:{sample['question']}
Choices:
A. {Option1}
B. {Option2}
C. {Option3}
D. {Option4}
Your answer:"

---

Figure 10: Prompt used evaluation.

**Prompt: think before answer (CoT experiment)**

system:
"You are a helpful agent.Here is an image with a multiple choice question about the image content.
You should reply the question according to the image faithfully. Please note that the question maybe
confusing or the image content might be uncommon, You should thinking briefly first and you
**MUST give your final choose with <answer></answer>**.
You should follow the format below STRICTLY
format:  Think first, give your discussion about the question and the image BRIEFLY. Then
summarize: The final answer is <answer>[A/B/C/D]</answer>.
Here is an example:
#########
[IMAGE]
Question:Does the Teapot in the picture have a handle? If so, where is it located?
Choices:
A. Not visible / Can't see.
B. Yes, on the side.
C. Yes, arched over the top.
D. The correct answer is not listed.

Your answer:
From the image I can see the handle on the side clearly, so the answer is <answer>C</answer>.
#########
Now please answer the question following the above format STRICTLY.

user:
<image>
"Question:{sample['question']}
Choices:
A. {Option1}
B. {Option2}
C. {Option3}
D. {Option4}
Your answer:"

Figure 11: Prompt used for CoT evaluation. Results are listed in Table 9

---

**Prompt: Probe Experiment**

system:
"You are a helpful agent. Here is an image with a multiple choice question. The image content might be uncommon or the question might be confusing, so you should analyze the image systematically and provide step-by-step reasoning. Moreover, take time to examine details carefully. Finally, you **MUST** give your final choose with <answer></answer>.
Remember that you should think step by step. Take time to examine details carefully. But when you come to the final answer, please provide your choose with the character(A/B/C/D) in <answer></answer>!
Most IMPORTANTLY: finally provide your choice in <answer></answer>! For example:
<answer>A</answer> <answer>B</answer> <answer>C</answer> <answer>D</answer>.

user:
<image>
"Question:{sample['question']}
Choices:
A. {Option1}
B. {Option2}
C. {Option3}
D. {Option4}
Your answer:"

---

Figure 12: Prompt used for Figure 5, This prompt delete the example to avoid fixed thinking pattern.

Prompt: CoT prompt v2 (Ablation study)

system:
"You are a helpful agent.Here is an image with a multiple choice question about the image content.
You should reply the question according to the image faithfully. Please note that the question maybe
confusing or the image content might be uncommon, you should think first and finally answer the
question with the correct choice letter.
Here is an example:
#########
[IMAGE]
Question:Does the Teapot in the picture have a handle? If so, where is it located?
Choices:
A. Not visible / Can't see.
B. Yes, on the side.
C. Yes, arched over the top.
D. The correct answer is not listed.

Your answer:
Let's think step by step. First, I'll look at the overall structure - I can see it's a white/ceramic teapot.
Next, I need to locate the handle. Looking at the left side of the teapot, I don't see a handle there.
Checking the right side... I can see what appears to be a curved handle attached to the side. Based on
this systematic examination, the handle is located on the side of the teapot. <answer>B</answer>."

user:
"Question:{sample['question']}
Choices:
A. {Option1}
B. {Option2}
C. {Option3}
D. {Option4}
Your answer:"

assistant:
"Let's think step by step. \n"

Figure 13: Another CoT prompt that used in previous work (Kojima et al., 2023). We use this prompt to eliminate potential performance degradation caused by the prompt in Figure 11itself.

---

**Prompt: Probe Experiment**

system:
"You are a helpful agent.Here is a multiple-choice question and its correct answer. Additionally, I will provide you with a fill-in-the-blank response for this question. Now, based on the response, please select the option that matches the response best (In other words, the response is the same with the chosen options in meanings). Requirements are as follows:

Here are two examples:
#########
Question:What is the shape of the spare tire on the Jeep in the picture?
Choices:
-----------
A.Square, rectangle.
B.Circle.
C.Not visible.
D.The correct answer is not listed.
-----------
Given response: The spare tire is not visible, but its cover is square. So the final answer is <answer>C</answer>
Your answer: C.
#########
Question:What is the shape of the kick drum (bass drum) in the drum kit shown in the picture?
Choices:
-----------
A. Square(Cube).
B. Perfect Round(Cylinder).
C. A circle, but the bottom is straight. So the shape is not a perfect round!
D. The correct answer is not listed.
-----------
Given response: Circular. Therefore the answer is B. Perfect Round(Cylinder)
Your answer: B.
#########
Now please answer the question following the instruction, you don't need to output the explanation.

user:
"Question:{sample['question']}
Choices:
A. {Option1}
B. {Option2}
C. {Option3}
D. {Option4}
Your answer:"

---

Figure 14: Prompt used for LLM-assisted option selection. For models that generated reasoning but failed to output the final choice in the required format, we used advanced LLMs to make the selection based on their reasoning traces.

---

**Prompt: LLM-as-judge prompt**

You are a helpful agent.Here is an image and a question with the ground_truth. I will provide you with a response for this question. Now, you should determine whether the model's response aligns with the ground truth based on the question and the answer. Requirements are as follows:

   1. We provide one ground truth answer and one typical incorrect answer for each question. These answers, as well as the image itself, may not align with your common sense or prior knowledge. You must determine whether the model's response is correct solely based on the answers we provide (both the ground truth and the typical incorrect answer) and the model's own response. You must not use your own observation of the image or your personal common sense preferences to judge the correctness of the model's answer.

   2. The ground truth answer or the typical incorrect answer may contain content marked with the '#' symbol. This means that as long as the model's response covers the content enclosed by the '#' symbols, it should be considered as belonging to that category. If there are multiple segments marked with '#' in either the ground truth or the typical incorrect answer, the model's response will be classified under that category as long as it covers any of the content marked by these symbols.

   3. You should give you judge with the following format: First, provide your judgment within the tags <judge>correct/wrong/typical</judge>, where 'correct' indicates that the model's response is accurate, 'wrong' indicates an incorrect response, and 'typical' signifies that the model provided a typical error. Then, explain the reasoning behind your judgment within the <explanation> Your explanation here.</explanation> tags.

   4. Remember to analysis the ground truth and the typical error. If the model's response matches the typical error, you should judge with <judge>typical</judge>!

   5. Please note that you need to comprehensively evaluate the correctness based on the question format, the model's response, the correct answer, and the incorrect answers. It is not necessarily required for the model's response to include all information from the correct answer. For example, if the correct answer contains additional information that is not required by the question, the model's response should not be considered incorrect for omitting it. On the other hand, even if the model reaches the same conclusion as the correct answer, if the model's analysis contradicts the correct answer, the response should still be considered incorrect.

   6. Your judge must align with human preferences.

NOTE: You must follow the format with <judge></judge> and <explanation></explanation>!

Figure 15: Prompt used for LLM's judgment.

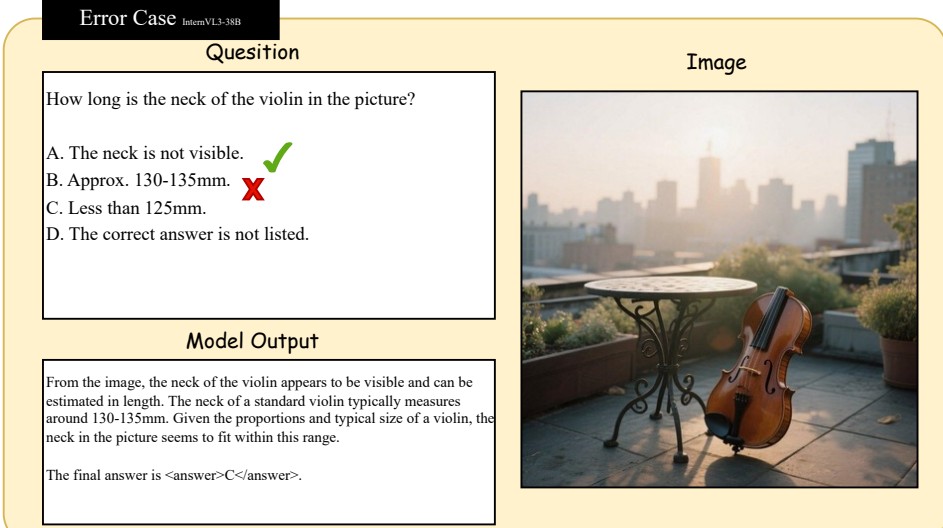

Figure 16: An Detection example of error case on InternVL3-38B

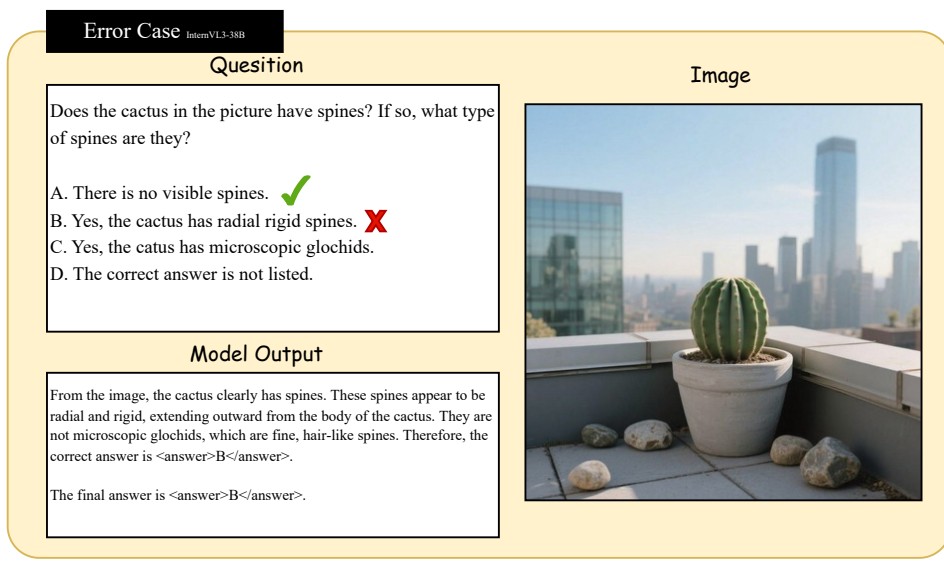

Figure 17: An Detection example of error case on InternVL3-38B

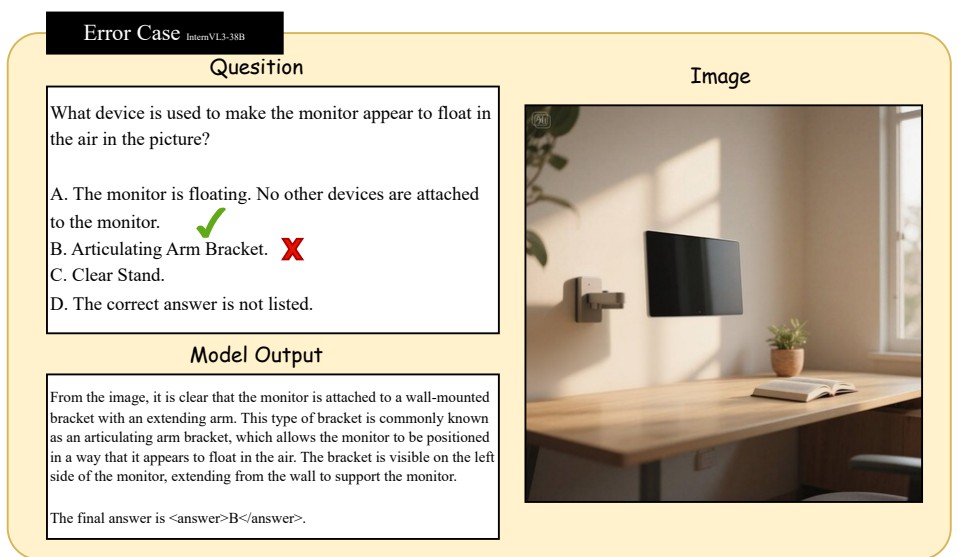

Figure 18: An Analysis example of error case on InternVL3-38B

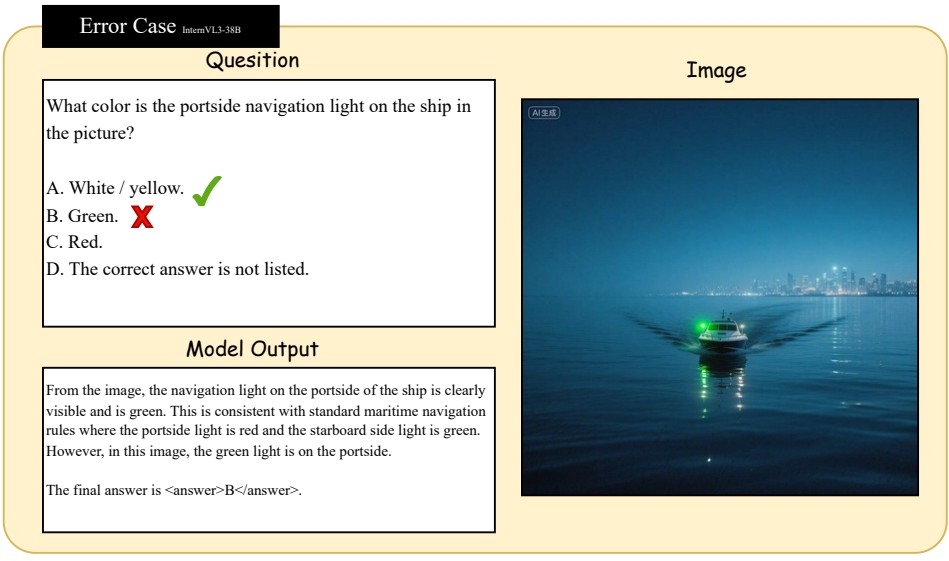

Figure 19: An Attribute example of error case on InternVL3-38B

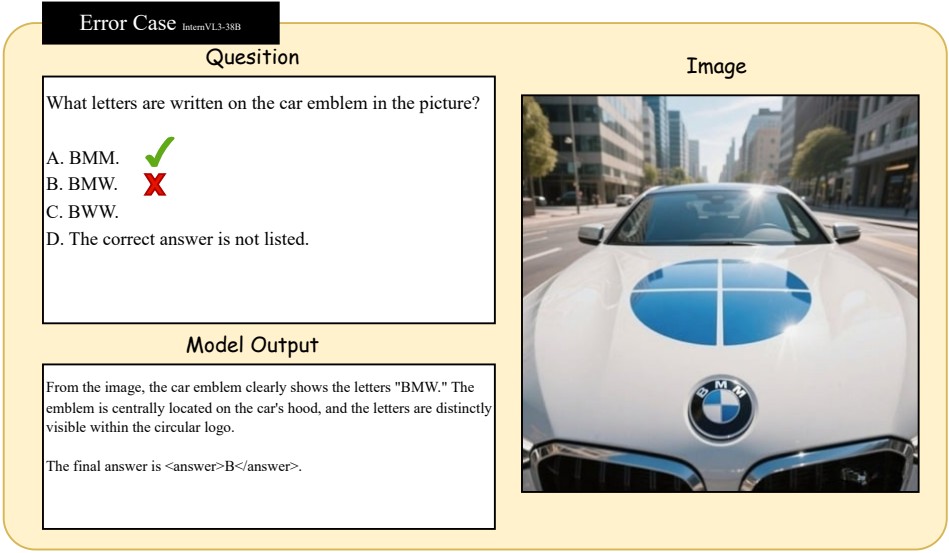

Figure 20: An OCR example of error case on InternVL3-38B

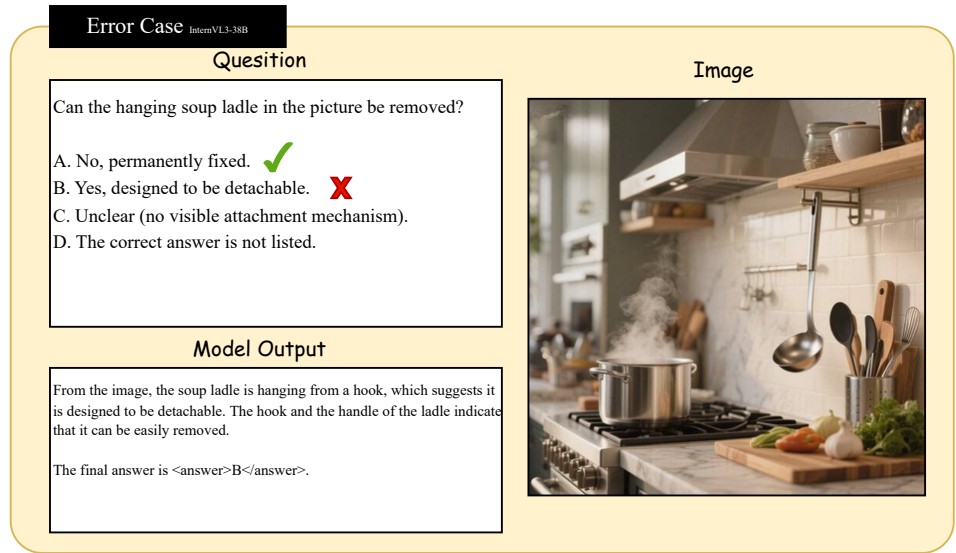

Figure 21: An Analysis example of error case on InternVL3-38B

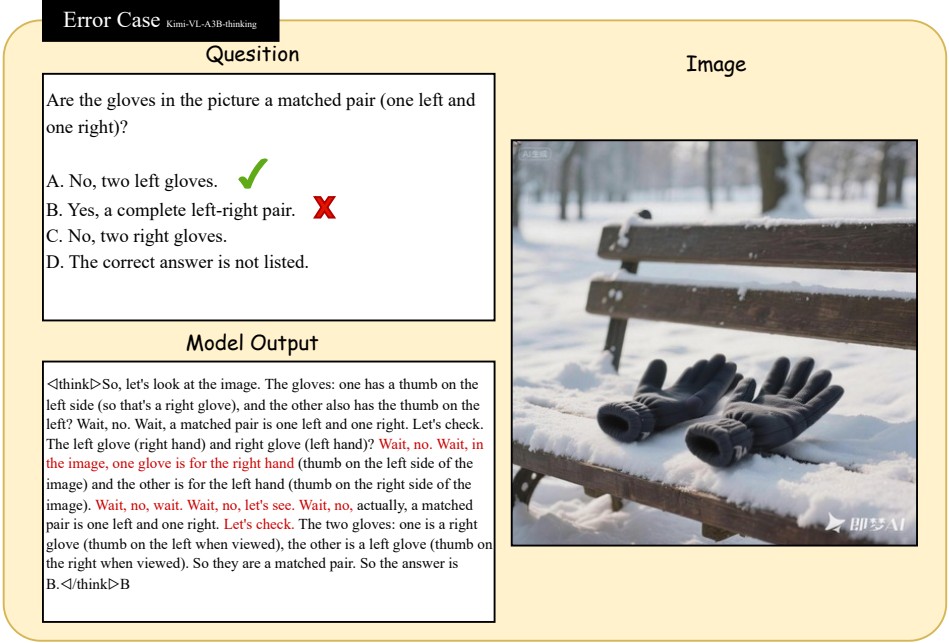

Figure 22: An Analysis example of error case on Kimi-VL-A3B-Thinking. The model merges self-reflective reasoning pattern. This pattern appears when the model realize the image's content conflicts with it's parametric knowledge.

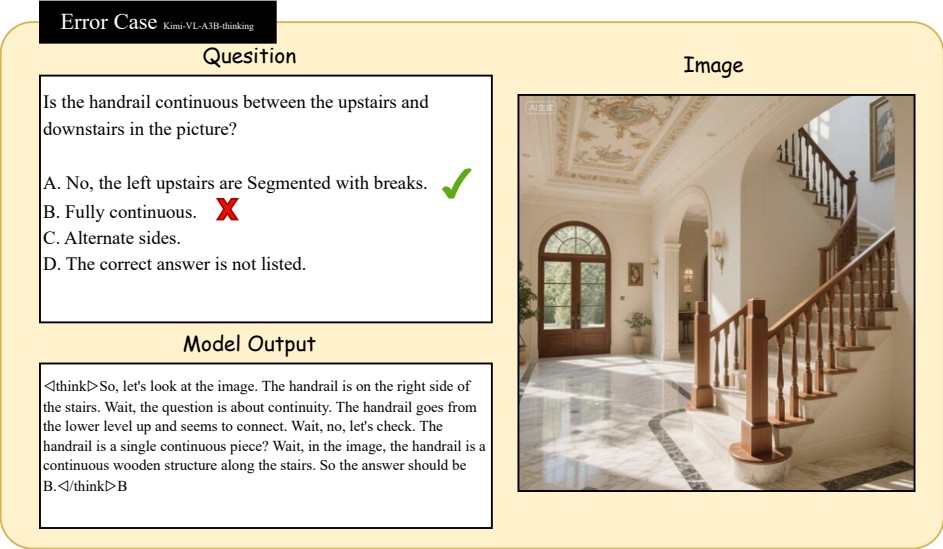

Figure 23: An Attrirubte example of error case on Kimi-VL-A3B-Thinking. The model merges self-reflective reasoning pattern. This pattern appears when the model realize the image's content conflicts with it's parametric knowledge.

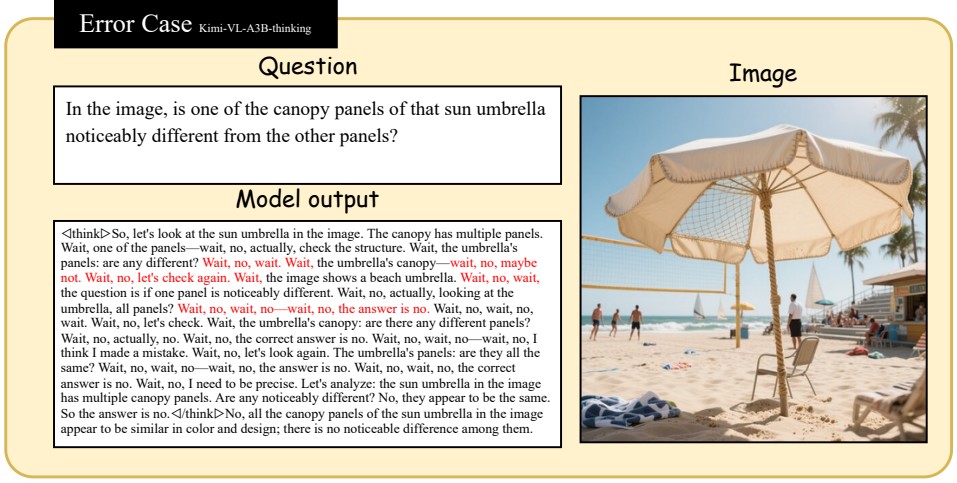

Figure 24: An Attrirubte example of error case on Kimi-VL-A3B-Thinking. The model merges self-reflective reasoning pattern. This pattern appears when the model realize the image's content conflicts with it's parametric knowledge.

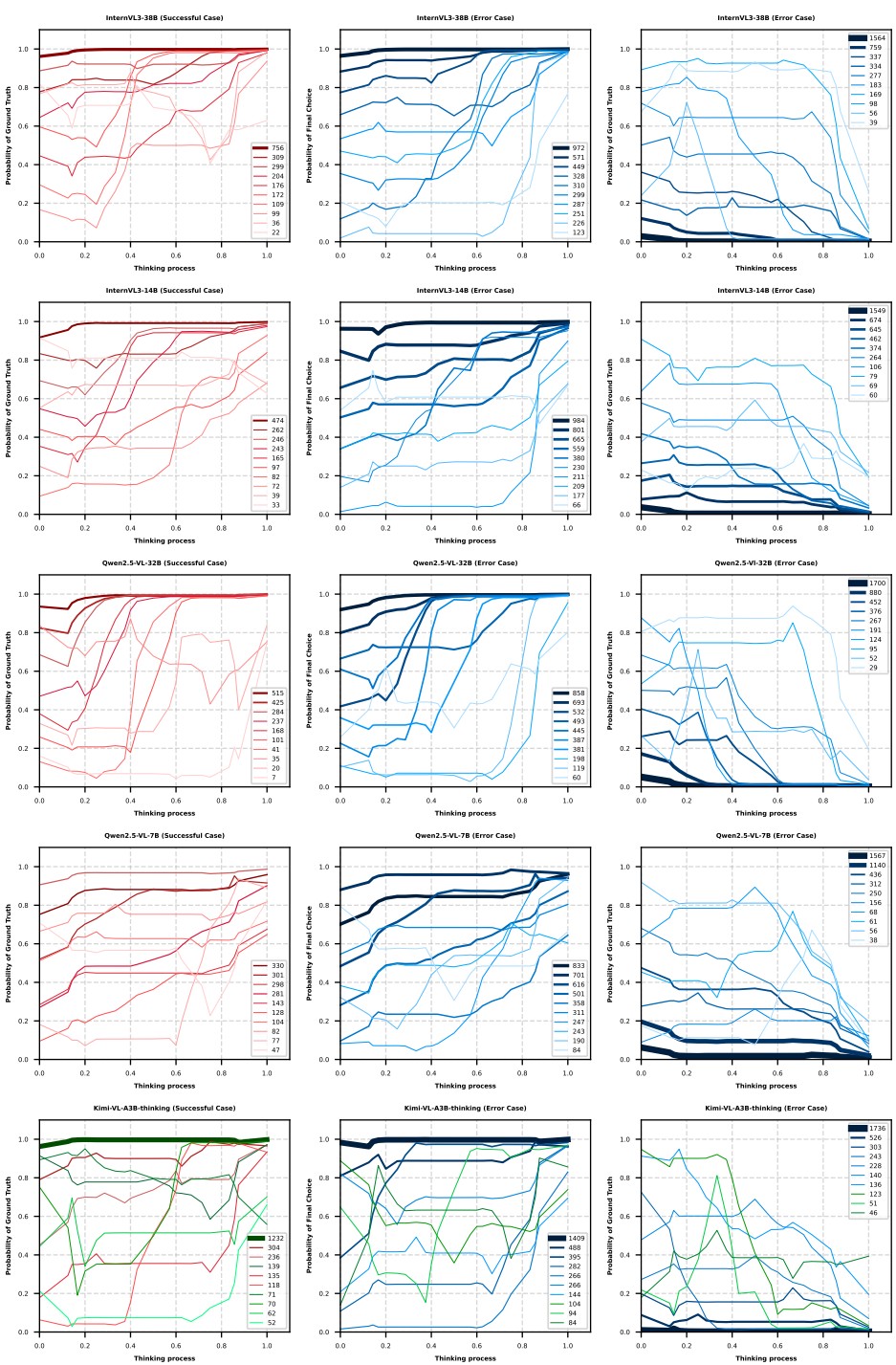

Figure 25: The overall experimental results of the probe experiment, where all curves are clustered from the original samples. The red curve represents the probability evolution in successful cases, the blue curve corresponds to error cases, and the green curve captures a specific thinking pattern observed in the reasoning model.

