# OpenReview forum: "FREAK: A Fine-grained Hallucination Evaluation Benchmark for Advanced MLLMs"
_ICLR.cc/2026/Conference — ICLR 2026 Poster_

### Official Review · Reviewer_Xkgu · 2025-10-27

**Soundness:** 2
**Presentation:** 3
**Contribution:** 3
**Rating:** 6
**Confidence:** 4

**Summary:**

This paper introduces FREAK, a benchmark for evaluating fine-grained hallucinations in MLLMs using photorealistic images with subtle counter-commonsense (CCS) edits created via a novel "generate-then-edit" pipeline. FREAK tests detailed perception across 6 categories. Experiments show SOTA MLLMs struggle significantly (~45% accuracy vs. 86.71% human), revealing severe limitations, and Chain-of-Thought prompting often degrades performance.

**Strengths:**

1. The paper writes well.
2. The focus on fine-grained hallucinations, where models find it difficult to perceive subtle details contradicting common sense, is highly relevant.
3. Instead of relying solely on direct text-to-image generation or manual editing, the two-stage process (generate normal -> edit CCS detail) allows for the creation of high-quality, photorealistic images with precisely controlled, localized CCS elements.
4. The gap between SOTA models and humans clearly indicates that fine-grained perception remains a major challenge.
5. The investigation into why CoT prompting degrades performance is valuable.

**Weaknesses:**

1. While the images are generated/edited, the initial CCS descriptions and the subsequent questions/distractors are generated by LLMs. This introduces the possibility of biases inherent in the LLMs used during benchmark creation.
2. The quality and nature of the CCS images depend heavily on the capabilities of Seedream3.0 and SeedEdit3.0.
3. It focuses primarily on perception, less on deeper reasoning.
4. The pipeline is model-assisted, but the dataset size is still very small. It would be better if it could be scaled up.

**Questions:**

1. The image editing process could introduce subtle visual artifacts. How do the authors ensure that models are identifying the semantic, counter-commonsense errors, rather than simply learning to detect these low-level editing artifacts?
2. The benchmark is explicitly designed for 'fine-grained perception.' While valuable, this focus seems to neglect hallucinations that arise from multi-step reasoning. Could the authors discuss this trade-off and the benchmark's limitations in assessing these more complex failure modes?
3. For a semi-automated pipeline, the resulting dataset of ~1.8k images is smaller than expected. Could the authors quantify the main bottleneck in their workflow (e.g., image generation failure rate, human verification time) that limits the benchmark's scalability?

---

> ### Author Response · Authors · 2025-11-20
> **Response to Reviewer Xkgu (1/3)**
>
> We thank the reviewer for the detailed and useful comments. We hope to address any concerns and questions below.
>
> ## **Weaknesses**
> ***
> **W1: LLM Bias**
> >1. While the images are generated/edited, the initial CCS descriptions and the subsequent questions/distractors are generated by LLMs. This introduces the possibility of biases inherent in the LLMs used during benchmark creation.
>
>
> We thank the reviewer for pointing out potential LLM-induced biases. We have taken multiple measures to minimize such biases in FREAK:
> 1. **Using multiple LLMs to generate CCS descriptions**\
> We observed that different models produce varied CCS descriptions for the same object. By leveraging this diversity, we aimed to make the CCS content as broad and diverse as possible, thereby avoiding biases that could arise from the repeated use of specific CCS phenomena.
> 2. **Manual review of all items**\
>  Although the dataset is entirely synthetically generated, our team conducted individual reviews of each sample—including images, questions, and answers—to ensure that FREAK is fair, reliable, and suitable for trusted use within the community.
> 3. **Validation through human experiments**\
>  We designed a set of guidelines to conduct unbiased human evaluations. The strong performance of human participants on FREAK demonstrates that its question design aligns with human reasoning and values, indirectly confirming that FREAK contains minimal or no LLM-induced biases.
>
> Together, these steps ensure that FREAK provides a reliable, high-quality benchmark for fine-grained hallucination evaluation, offering improvements in scale, diversity, and rigor compared to prior CCS-style evaluations.
> ***
> **W2: Image Quality**
> >The quality and nature of the CCS images depend heavily on the capabilities of Seedream3.0 and SeedEdit3.0.
>
> We believe that the emergence of advanced generation/editing models such as Seedream and Nano Banana is precisely what enabled the construction of FREAK. Our proposed generate-then-edit pipeline is deliberately simple, effective and model-agnostic. As the capabilities of generative and editing models continue to advance in the future, this pipeline will remain effective and is expected to produce outputs of increasingly higher quality, which we consider one of our key contributions.
>
> Furthermore, to assess potential biases introduced by SeedEdit 3.0, we conducted an ablation experiment focusing on image editing models. We randomly sampled 114 instances and modified them using different editing models. The results show that the accuracy across SOTA models remained stable, with answer consistency exceeding 80%. This experiment verifies that using SeedEdit 3 does not introduce structural bias, thereby demonstrating the reliability of the FREAK dataset.
>
> | Model               | SeedEdit 3.0 Accuracy | SeedEdit 3.0 Consistency | Nano Banana Accuracy | Nano Banana Consistency | Seedream 4.0 Accuracy | Seedream 4.0 Consistency |
> |---------------------|----------------------|-------------------------|---------------------|-------------------------|----------------------|-------------------------|
> | Gemini2.5 flash     | 46.23%               | -                       | 44.78%              | 83.18%                  | 47.54%               | 79.99%                  |
> | o4 mini             | 45.86%               | -                       | 44.20%              | 83.04%                  | 50.14%               | 83.48%                  |
> | Qwen2.5-VL 32B      | 38.26%               | -                       | 38.12%              | 82.32%                  | 39.71%               | 86.96%                |
> ***
> **W3: Missing Reasoning Hallucination**
> >It focuses primarily on perception, less on deeper reasoning.
>
> We thank the reviewer for this comment. **Perception capability** is a foundational competency of multimodal models. If an MLLM cannot reliably perceive visual content, its reasoning outputs will inherently lack grounding in reality. Evaluation results on FREAK demonstrate that **current MLLMs’ perception is not yet reliable, highlighting the importance of establishing an advanced hallucination benchmark**, which is the primary focus of FREAK.
>
> While deeper reasoning is indeed important, it is more closely associated with LLMs and goes beyond the core focus of FREAK. We acknowledge that exploring reasoning-related hallucinations in MLLMs is a valuable direction and plan to investigate it further in future work.

---

> > ### Author Response · Authors · 2025-11-20
> > **Response to Reviewer Xkgu (3/3)**
> >
> > **Q3: Data Size and Scalability**
> > > For a semi-automated pipeline, the resulting dataset of ~1.8k images is smaller than expected. Could the authors quantify the main bottleneck in their workflow (e.g., image generation failure rate, human verification time) that limits the benchmark's scalability?
> >
> > Our semi-automated pipeline demonstrates significant advantages over existing methods for generating CCS images in terms of both quality and quantity. However, during the construction of FREAK, we observed certain efficiency limitations in the current pipeline. The primary bottlenecks restricting the scale of FREAK are as follows:
> >
> > 1. **Low quality of CCS descriptions generated by LLMs.**\
> > The main issue is repetition: for similar entities, the LLM often produces very similar CCS descriptions. And the secondary issue is that some outputs fail to strictly satisfy the “counter-common-sense only in details” requirement.\
> > Despite few-shot guidance, low-quality descriptions still occur. From a random sample of 1,000 filtered low-quality items, we found that **~79% (787 instances)** were discarded due to low-quality CCS descriptions, while **~21% (213 instances)** were discarded because the generated image did not match an otherwise acceptable description. This indicates that the primary bottleneck is the LLM’s inefficiency in generating sufficiently diverse and precise CCS descriptions.
> >
> > 2. **Instability in image generation.**\
> > Some CCS descriptions require multiple invocations of the generative/editing models to produce a valid image that matches the description. This adds to the time and computational cost.
> >
> > 3. **High cost of manual verification.**\
> > To ensure quality, all items underwent careful human review. This step is necessary due to the issues above, and it naturally constrains the overall dataset size.
> >
> > For the relatively low efficiency of LLMs, we analyse this is because the **LLM has no memory across sessions, so it tends to repeat similar descriptions for similar entities, even with few-shot prompts**.
> >
> > In summary, the main bottleneck is the low quality of LLM-generated CCS descriptions, followed by image generation instability and manual verification costs. Our semi-automated pipeline, however, provides a foundation for future scaling once these challenges are addressed.
> >
> > ***
> > We hope the above responses address your concerns and provide a clearer understanding of our manuscript. We would greatly appreciate it if you could consider increasing your rating.

---

> ### Author Response · Authors · 2025-11-20
> **Response to Reviewer Xkgu (2/3)**
>
> **W4: Scaling Ability**
> >The pipeline is model-assisted, but the dataset size is still very small. It would be better if it could be scaled up.
>
> We appreciate the reviewer’s concern regarding dataset size. While FREAK contains 1.8K items, we believe this scale is appropriate and sufficient at the current stage, for four reasons:
>
> 1. **Quality over quantity.**\
> To avoid evaluating MLLMs on repetitive or redundant CCS phenomena, we aggressively filtered out low-quality or semantically duplicated CCS descriptions. This deliberate curation ensures that each item tests a distinct fine-grained capability.
>
> 2. **High manual verification cost.**\
> Roughly 80% of the initially generated samples were discarded after human inspection due to repetitive descriptions, low-quality images, or mismatches. Every retained item was manually verified, which makes uncontrolled scaling infeasible at present.
>
> 3. **The pipeline itself is designed for future expansion.**\
> Our semi-automated pipeline requires only a list of entities and can generate CCS descriptions and images automatically. Thus, FREAK can be expanded easily in future work once the cost of human verification is reduced.
>
> 4. **Competitive scale relative to prior CCS-style benchmarks.**
> Existing datasets (e.g., HallusionBench (CVPR2024, 436 imgs), WHOOPS (ICCV2023, 500 imgs), and PhD-ccs(CVPR2025, 750 imgs)) remain small due to their generation methodologies. At 1.8K items, FREAK already represents a substantial improvement while maintaining high quality.
>
> In summary, although the dataset could be expanded in the future, the current size reflects a deliberate balance between quality, manual verification cost, and fairness. We believe 1.8K high-quality, fine-grained examples are adequate for reliably evaluating modern MLLMs.
> ***
> ## **Questions**
> ***
> **Q1: Editing Artifacts**
> > The image editing process could introduce subtle visual artifacts. How do the authors ensure that models are identifying the semantic, counter-commonsense errors, rather than simply learning to detect these low-level editing artifacts?
>
> Our task design prevents models from exploiting low-level artifacts. All questions in FREAK require high-level semantic understanding (e.g., recognizing an object and identifying its counter-common-sense modification), so answering correctly necessarily reflects semantic comprehension rather than artifact detection.
>
>  In addition, every image was manually verified to ensure visual coherence without noticeable editing traces.
>
> We also include free-form questions, where correctness depends on producing semantically consistent descriptions, further discouraging artifact-based shortcuts.
>
> In our response to **Weakness 2**, we introduced an ablation experiment involving different editing models. The results confirm that SeedEdit 3 does not introduce structural bias specific to that editor. The stable performance observed across multiple editing models indicates that model performance is not influenced by artifacts unique to any particular editing method.
> ***
> **Q2: Limitation in Complex Failure Modes**
> >The benchmark is explicitly designed for 'fine-grained perception.' While valuable, this focus seems to neglect hallucinations that arise from multi-step reasoning. Could the authors discuss this trade-off and the benchmark's limitations in assessing these more complex failure modes?
>
> As stated in response to **Weakness 3**, FREAK is intentionally designed to isolate fine-grained visual hallucinations. Our goal is to evaluate whether MLLMs can accurately perceive subtle, counter-common-sense details in images, rather than hallucinations that emerge from long-chain or purely textual reasoning. To ensure fairness across models with different reasoning capabilities, we adopt straightforward, perception-oriented questions that rely minimally on multi-step inference and remain easily answerable by humans.
>
> We fully acknowledge that multimodal hallucinations induced by multi-step reasoning are important and increasingly relevant with the rise of reasoning-centric MLLMs. This is a distinct failure mode and an exciting direction for future benchmarks, but it lies outside the intended scope of FREAK.

---

### Official Review · Reviewer_bnQv · 2025-10-30

**Soundness:** 4
**Presentation:** 3
**Contribution:** 4
**Rating:** 6
**Confidence:** 4

**Summary:**

This paper proposed FREAK, a challenging hallucination benchmark for vision-language models. It follows a 3-step pipeline at construction: 1) generate counter-commonsense (CCS) textual description using LLMs 2) generate CCS images using an image editing models and 3) generate MCQ/open-ended questions using LLMs followed by postprocessing and a human inspection process. Results show that even humans only achieve 86% accuracy, followed by gemini-2.5-pro at 43%, indicating the challenging nature of this benchmark. They also find that CoT/reasoning does not bring performance boost.

**Strengths:**

1. The benchmark dataset is developed with strong motivation and should be beneficial for diagnosing models.

2. The blind-test human experiments for  validation makes the dataset more solid.

3. The in-depth analysis in sec6 provides insights on why the model hallucinates, potentially inspiring better model development.

**Weaknesses:**

1. A major drawback of this work is the taxonomy is not exhaustive or mutually exclusive, nor is it hierarchical. For instance, in fig2, the example question for the “Analysis” dimension also seems to fit into the “Position” dimension. Besides, while spatial relation is included, questions to probe general relations between objects seem missing (e.g. the “riding” relation as in “a person riding a horse”).

2. In sec4.2, why are errors divided into commonsense errors and others? Meanwhile, using LLMs to classify error types is not validated.

3. In sec4.2, “Hallucination Rate (HalluRate) is the proportion of cases where the model either outputs a commonsense answer in free-form questions or selects the commonsense distractor in multiple-choice questions.” How to judge whether a model outputs a commonsense answer? Is every open-ended question paired with a commonsense answer for judgement?

4. In sec6.1, can you elaborate on the input/output/groundtruth of the normal and CCS settings?

**Questions:**

1. Tab1 can be better formatted. Also, the fourth column should be “GPT Series Eval”

2. In line 243-244, when multiple groundtruths may exist, why is the dominant distractor adopted to construct the dataset?

3. One of the goals of blind-test human experiment is to detect potential biases, whose results are not shown. Did you find any bias in the annotated datasets?

4. It’s better to provide some analysis on why reasoning does not boost performance.

---

> ### Author Response · Authors · 2025-11-20
> **Response to Reviewer bnQv (1/3)**
>
> We thank the reviewer for the detailed and useful comments. We hope to address any concerns and questions below.
> ## **Weaknesses**
> ***
> **W1: Taxonomy**
> >A major drawback of this work is the taxonomy is not exhaustive or mutually exclusive, nor is it hierarchical ...
>
> We thank the reviewer for the feedback on our taxonomy design. We acknowledge that the original taxonomy had limitations, such as overlapping categories (e.g., some “Analysis” questions also involve “Position”). Please allow us to clarify that the primary intention behind designing multiple categories was to enable a more fine-grained analysis of MLLMs' performance in handling different types of problems.
>
> To address this, we revised the "taxonomy" framework into a **subtask-based design** with refined definitions, where each subtask targets a specific capability of MLLMs. We believe this revised structure more accurately conveys our objectives and avoids potential misinterpretations. Once again, we express our gratitude for your insightful suggestions.
> ***
> **W2: Commonsense Error Issue**
> >In sec4.2, why are errors divided into commonsense errors and others? Meanwhile, using LLMs to classify error types is not validated.
>
> In FREAK, errors are divided into commonsense-driven and other errors to reveal whether a model’s failure is caused by overreliance on linguistic priors (i.e., hallucination) or by genuine visual-perception limitations. CCS images naturally afford this distinction because each question has both a visually correct answer and a corresponding commonsense answer.
>
> The two error types differ in whether the model has effectively used the visual signal:
> - **Commonsense error**: the model ignores the image and answers according to its internal knowledge priors.
> - **Other error**: the vision or alignment module fails to provide sufficient visual detail, and the model’s answer aligns with neither the ground truth nor the commonsense distractor.
>
> To categorize model outputs, we provide the ground-truth answer and the commonsense answer to the judge LLM and ask it to determine whether the model’s response has the same semantic meaning with groundtruth or commonsense answer. This procedure does not require the judge to analyse deeply; it only needs to compare MLLM's output between two explicit candidates.
> Regarding reliability, we validated the LLM-as-judge method against human annotations (Appendix C.1.3). The consistency exceeds 90% across multiple models, demonstrating that the judge can reliably distinguish between these two predefined error types.
> ***
> **W3: Judgement of Commonsense Answers**
> >In sec4.2, “Hallucination Rate (HalluRate) is the proportion of cases where the model either outputs a commonsense answer in free-form questions or selects the commonsense distractor in multiple-choice questions.” How to judge whether a model outputs a commonsense answer? Is every open-ended question paired with a commonsense answer for judgement?
>
> Yes! Every question in FREAK includes both the correct answer and a commonsense answer, which is a distinctive feature of its design.
> - For multiple-choice items, each question includes the correct option alongside a carefully designed commonsense distractor specific to both the question and its corresponding image.
> - For free-form questions, we supplemented the annotation process by additionally providing a commonsense answer. This commonsense answer is determined based on the commonsense image content before it was modified into a CCS image.
>
> By incorporating these designed commonsense answers, we enable the direct and systematic quantification of the model's hallucination rate using the HalluRate metric.
> ***
> **W4: Subset Setting**
> >4. In sec6.1, can you elaborate on the input/output/groundtruth of the normal and CCS settings?
>
> In Sec 6.1, each CCS image ($CCS$) in FREAK is uniquely paired with a normal image ($I$), both corresponding to the same question ($Q$) and answer choices ($[X,Y,Z,W]$).
> - **Input**: For each group, the model receives the image, question, and options as a prompt.
> - **Output**: The model selects an answer.
> - **Ground-truth**: For $CCS$, the correct answer is the ground turth as provided in FREAK; for $I$, the commonsense distractor of $CCS$ is exactly the correct answer.
>
> For example, as we showed in Fig.6, the CCS image shows a fruit knife blade standing vertically while the handle lies flat, violating common knowledge. The normal image represents the original state. Through this controlled comparison where only the image varies, we aim to indirectly validate that:
> 1. Even if the model selects an incorrect answer for the $CCS$ image, it may still extract correct semantic information (for related analysis, see Sec. 6.1 of the paper).
> 2. While the model performs well on the original image $I$, its performance drops significantly when evaluated on the CCS image, indicating severe hallucinations in fine-grained visual perception.

---

> ### Author Response · Authors · 2025-11-20
> **Response to Reviewer bnQv (2/3)**
>
> ## **Questions**
> ***
> **Q1: Table 1 format**
> >Tab1 can be better formatted. Also, the fourth column should be “GPT Series Eval”
>
> We thank you for your valuable feedback. We have revisited the table headers and revised the potentially misleading phrasing you pointed out.
> ***
> **Q2: Dominant Distractor**
> >In line 243-244, when multiple groundtruths may exist, why is the dominant distractor adopted to construct the dataset?
>
> When multiple plausible commonsense answers exist for a question, we adopt **the dominant distractor** to ensure consistency and maintain credible evaluation. For instance, if an edited image shows a guitar with only five strings, the pre-edit state could correspond to a standard six-string guitar or a four-string ukulele. In such cases, we select the most typical real-world scenario as the commonsense distractor, as MLLMs are trained on more data with these typical scenarios than others.
>
> This unified choice simplifies metric HalluRate calculation, while preserving the validity of the annotations across the benchmark. Even in these edge cases, the approach ensures systematic and reliable evaluation of model hallucinations.
> ***
> **Q3: Bias of Dataset**
> >One of the goals of blind-test human experiment is to detect potential biases, whose results are not shown. Did you find any bias in the annotated datasets?
>
> The blind-test human experiment aimed to verify the objectivity and fairness of FREAK annotations. And **no significant bias was detected**: human participants performed consistently well across the subset of questions, indicating that the images, questions, and answer choices align with human reasoning.
>
> To further minimize bias, each sample in FREAK underwent a rigorous review process, and annotation tasks were randomly assigned among team members to prevent individual bias. This validates the reliability of the annotated dataset.
> ***
> **Q4: Analysis of Reasoning Process**
> > It’s better to provide some analysis on why reasoning does not boost performance.
>
> Our analysis indicates that reasoning strategies do not improve performance because, **even when the model correctly extracts visual information, the extended reasoning process can override it with outputs influenced by internal commonsense knowledge**.
>
> Specifically, during early to middle reasoning stages, the model’s probability for the correct answer remains stable (a plateau phase), but in mid-to-late stages, it often declines sharply as the model generates text contradicting the initial correct interpretation, ultimately producing a hallucinated output. We interpret this as a form of self-doubt: the model attempts to **“correct” its initially accurate answer to align with pre-learned knowledge**, but this process instead introduces errors.
>
> Here is an example of Kimi-VL-A3B-Thinking's wrong thinking process in an error case (Fig. 24 in the paper).
>
> >Are the gloves in the picture a matched pair (one left and  one right)?\
> A. No, two left gloves.  (Correct Answer)\
> B. Yes, a complete left-right pair.  (Commonsense Distractor)\
> C. No, two right gloves.\
> D. The correct answer is not listed.\
> **Model output**: ◁think▷So, let's look at the image. The gloves: **one has a thumb on the left side (so that's a right glove), and the other also has the thumb on the left?** (*This is the correct description of the image*) **Wait, no. Wait**, a matched pair is one left and one right. **Let's check**. The left glove (right hand) and right glove (left hand)? Wait, no. Wait, in  the image, **one glove is for the right hand (thumb on the left side of the  image) and the other is for the left hand (thumb on the right side of the  image)** (*This description has already contradict the output in previous stages, which cause model's error final answer*). Wait, no, wait. Wait, no, let's see. Wait, no, actually, a matched  pair is one left and one right. Let's check. The two gloves: one is a right  glove (thumb on the left when viewed), the other is a left glove (thumb on  the right when viewed). So they are a matched pair. So the answer is  B.◁/think▷B
>
> The similar patterns are observed in other reasoning models such as MiMO-VL-RL and Skywork-R3. In contrast, non-reasoning models without explicit self-refinement do not exhibit such prolonged trajectories, yet they still exhibit the same trend of thinking process.
>
> In summary, reasoning fails to enhance performance because extended thinking often amplifies hallucinations rather than correcting them.
>
> ***
> We hope the above responses address your concerns and provide a clearer understanding of our manuscript. We would greatly appreciate it if you could consider increasing your rating.

---

> > ### Comment · Reviewer_bnQv · 2025-11-25
> >
> > Thanks for your detailed reply which cleared out most of my questions. The analysis of reasoning is interesting and should be included in the updated manuscript.
> >
> >
> > Below are 3 issues remained:
> >
> > 1. The sub-task paradigm doesn't fully solve the issue that some categories overlap. One naive solution is to discard the "analysis" task since all other tasks require some analysis.
> >
> > 2. **Bias and limitation of the dataset**: "human participants performed consistently well across the subset of questions" doesn't mean there is no bias in the collected dataset. Bias can be systematic (e.g. a large proportion of samples containing more men than women). Also more limitations should be mentioned than just small sample size (line 509). For instance, since image editing models are used, there can be unintended edits/artifacts in resulted images. If possible, inspections on this can be performed to clear out this potential issue.
> >
> > 3. fig 16-24 are not referred to in the paper.

---

> ### Author Response · Authors · 2025-11-26
> **Response to Reviewer bnQv (3/3)**
>
> We sincerely appreciate your constructive feedback and are glad that our previous response addressed most of your concerns. Below, we further respond to the three remaining questions point-by-point.
> ***
> **W.a Regarding the sub-task paradigm and category overlap**
>
> Thank you for raising concerns about the rigor of the sub-task paradigm. We would like to clarify why we retain the Analysis sub-task and why the current design remains reasonable:
> - **Functional distinction of the Analysis sub-task**\
> As stated in the manuscript, our intention in designing multiple sub-tasks is to evaluate model capability across different reasoning settings. Unlike others, the Analysis sub-task does not explicitly point the model to the critical region or object in the image. For example, in Fig. 2, all other sub-tasks guide MLLMs to a specific object or attribute, requiring perception-oriented extraction. In contrast, the Analysis sub-task requires the model to independently locate visual cues and form reasoning, making it more inference-centric rather than perception-driven.
> - **Partial overlap between sub-tasks is natural and acceptable.**\
> As discussed in Line 287, the purpose of the sub-task design is to simulate a variety of problem types. Complete orthogonality is unrealistic for a VQA benchmark, and some conceptual overlap is inevitable. We believe this overlap does not weaken the benchmark but instead reflects the complexity of real-world application.
>
> ***
> **W.b Dataset bias and limitations**
>
> Thank you for pointing out the necessity of elaborating dataset bias beyond sample size. We agree with your suggestion and have updated the manuscript accordingly. We now provide a discussion of Limitations & Future Work in Sec. 7 (highlighted in blue).
>
> To further examine potential bias introduced by editing models, we conducted an additional ablation experiment to verify whether SeedEdit 3.0 may introduce imperceptible artifacts or systematic distortions. The results are presented in Appendix D.1 (highlighted in blue). And the key finding is that the accuracy of MLLMs are stable when using different editing models. These results indicate that images generated by different editing models do not materially affect evaluation outcomes, reducing concerns about unobserved artifacts or editing-induced bias.
> ***
> **W.c Missing citations for Figure 16–24**
>
> Thank you for your careful reading. We have revised the manuscript and added appropriate references to Figures 16–24 in Appendix E.

---

> > ### Comment · Reviewer_bnQv · 2025-11-27
> >
> > Thanks for your reply. Scores are updated.

---

> > > ### Author Response · Authors · 2025-11-28
> > > **Response to Reviewer bnQv**
> > >
> > > Thank you for the update and for your careful review.
> > >
> > > We sincerely appreciate your time and constructive feedback!

---

### Official Review · Reviewer_EXmZ · 2025-10-31

**Soundness:** 3
**Presentation:** 3
**Contribution:** 3
**Rating:** 6
**Confidence:** 2

**Summary:**

This paper introduces a new fine-grained hallucination benchmark for Multimodal Large Language Models (MLLMs), addressing the limitations of previous benchmarks, which often suffer from low quality, oversimplified tasks, saturated metrics, and limited diversity. The proposed benchmark is constructed from AI-generated, fine-grained counter-commonsense (CCS) images. To build it, the authors first define CCS concepts and then generate and refine images using a generative model. Experimental results show that current MLLMs still perform significantly worse than human experts on this benchmark. Furthermore, the study reveals that reasoning-based approaches such as Chain-of-Thought (CoT) can actually degrade performance in this setting.

**Strengths:**

* This paper highlights the limitations of previous MLLM hallucination benchmarks, such as low quality, oversimplified tasks, and insufficient diversity. The authors clearly compare their proposed benchmark, FREAK, with existing CCS benchmarks and demonstrate its superiority.

* The finding that Chain-of-Thought (CoT) reasoning negatively impacts performance on fine-grained hallucination tasks is insightful and well explained.

* The experiments are extensive, and the analyses are thorough and well-structured.

**Weaknesses:**

Major Weaknesses

* **Dependence on the Image Editing Model:** The quality of the generated CCS images likely depends heavily on the performance of the image editing model. Since this model is not trained specifically on CCS images, it may struggle to edit them properly, potentially leading to corrupted or unrealistic results. It is unclear how the authors ensured that the edited images appear natural. An ablation study or analysis regarding the influence of the image editing model on the benchmark quality would be valuable.

* **Insufficient Justification for the Fine-Grained Hallucination Benchmark:** The motivation for introducing a fine-grained hallucination benchmark is not well justified. The CCS images seem highly artificial, and it is questionable whether users would encounter such counter-commonsense images in real MLLM usage scenarios. The paper should provide stronger reasoning for why this type of benchmark is necessary and how it reflects practical hallucination issues.

* **Reliability of the LLM-as-Judge Evaluation:** The paper adopts an LLM-as-judge approach for free-form question evaluation, but its reliability is uncertain. It would strengthen the work to validate this approach by comparing its results with human judgments and analyzing the performance gap between LLM-based and human evaluations.

Minor Weaknesses

* **Ambiguity in Wording (Line 40–41):** The phrase “This saturation arises from inherent limitations in both difficulty and evaluation methods” is unclear. The authors should clarify what is meant by inherent limitations in this context.

* **Clarity in Table 2:** It would be helpful to separate reasoning-based models from non-reasoning models in Table 2\. Without this distinction, the statement “reasoning shows no clear advantage” does not align well with the presented results.

* **Typos and Formatting Issues**
  * Line 53: diversity the of → the diversity of
  * Line 141: judgments (spelling issue)
  * Line 147: MIRAGE Dong et al. (2025) (missing space)
  * Table 4: The arrow direction in the “Accuracy – CoT” column for GPT-4.1 appears incorrect.

**Questions:**

* How reliable is the image editing model used for generating CCS images? Are there any ablation studies or analyses that verify its impact on the benchmark quality?

* What is the clear rationale for introducing a fine-grained hallucination benchmark in MLLMs? The paper should better explain why such a benchmark is necessary and how it addresses real-world hallucination issues.

* How reliable is the LLM-as-judge approach used for evaluating free-form questions? Has its consistency with human evaluation been validated?

#####

---

> ### Author Response · Authors · 2025-11-20
> **Response to Reviewer EXmZ (1/3)**
>
> We thank the reviewer for the thoughtful comments. We hope to address any concerns and questions below.
>
> ## **Major Weakness & Questions**
> ***
> **W1 & Q1**
> >**Dependence on the Image Editing Model**\
> **Q:** How reliable is the image editing model used for generating CCS images? Are there any ablation studies or analyses that verify its impact on the benchmark quality?
>
> To assess whether the benchmark quality is overly dependent on SeedEdit 3.0, we conducted an ablation study using three different image-editing models: **SeedEdit 3.0**, **Nano Banana**, and **Seedream 4.0**. We randomly sampled 114 MCQ items from FREAK, applied the same editing prompt to all three models, and evaluated the resulting images using three representative MLLMs (Gemini-2.5-flash, o4-mini, and Qwen2.5-VL-32B).
>
> | Model               | SeedEdit 3.0 Accuracy | SeedEdit 3.0 Consistency | Nano Banana Accuracy | Nano Banana Consistency | Seedream 4.0 Accuracy | Seedream 4.0 Consistency |
> |---------------------|----------------------|-------------------------|---------------------|-------------------------|----------------------|-------------------------|
> | Gemini2.5 flash     | 46.23%               | -                       | 44.78%              | 83.18%                  | 47.54%               | 79.99%                  |
> | o4 mini             | 45.86%               | -                       | 44.20%              | 83.04%                  | 50.14%               | 83.48%                  |
> | Qwen2.5-VL 32B      | 38.26%               | -                       | 38.12%              | 82.32%                  | 39.71%               | 86.96%                |
>
> Across models, the answer consistency between SeedEdit 3.0 and the other editors ranges from 80% to 87%, while the accuracy differences for each MLLM remain small. These findings suggest that although different editors may vary slightly in CCS editing, the downstream reasoning difficulty remains largely stable. Therefore, benchmark performance is **not dominated or distorted by the specific choice of SeedEdit 3.0**, and potential information leakage or corruption unique to this editor can be ruled out.
>
> Importantly, during dataset construction, all edited images underwent **manual quality verification**, ensuring that only visually natural and semantically coherent images were included in the benchmark. This further mitigates concerns regarding artifacts introduced by any single editing model.
>
> Taken together, these results demonstrate that FREAK is robust to the choice of image-editing model, and SeedEdit 3.0 serves as a reliable editor for constructing CCS images.
> ***
> **W2 & Q2**
> >**Insufficient Justification for the Fine-Grained Hallucination Benchmark**\
> **Q:** What is the clear rationale for introducing a fine-grained hallucination benchmark in MLLMs? The paper should better explain why such a benchmark is necessary and how it addresses real-world hallucination issues.
>
> We thank the reviewer for raising this point. Our benchmark is not intended to reproduce the frequency of CCS-like images in real-world usage, but to diagnose a fundamental failure mode of MLLMs: **their inability to reliably perceive and reason about fine-grained object attributes even when object presence is correctly identified**.
>
> Existing hallucination benchmarks (e.g., POPE, AMBER) mainly target coarse-grained object existence errors. However, safety-critical failures in real applications often stem from incorrect attribute reasoning rather than object misdetection. Such fine-grained perception errors are widely reported in medical imaging[1], autonomous driving [2], and industrial inspection [3], where a single misinterpreted attribute can lead to severe consequences.
>
> Because fine-grained CCS cases rarely appear in natural datasets, it is difficult to evaluate whether a model genuinely understands subtle visual attributes or merely relies on priors. Following established CCS evaluation practice, we therefore construct minimally altered, localized CCS images that preserve global realism while modifying a single key attribute. This controlled design allows us to precisely test whether the model (a) perceives the attribute change, (b) updates its reasoning accordingly, and (c) avoids hallucinations inconsistent with the visual input.
>
> [1] A Survey of Multimodal Hallucination Evaluation and Detection, arxiv, 2507.19024\
> [2]  ScVLM: Enhancing Vision-Language Model for Safety-Critical Event Understanding, WACV2025\
> [3] EIAD: Explainable Industrial Anomaly Detection Via Multi-Modal Large Language Models, arxiv, 2503.14162

---

> ### Author Response · Authors · 2025-11-20
> **Response to Reviewer EXmZ (2/3)**
>
> **W3 & Q3**
> >**Reliability of the LLM-as-Judge Evaluation**\
> **Q:** How reliable is the LLM-as-judge approach used for evaluating free-form questions? Has its consistency with human evaluation been validated?
>
> In Appendix C.1.3, we provide a detailed comparison between human evaluations and LLM-as-judge results on free-form questions. Our setup evaluates the consistency of the judge model against human annotators across MLLMs of different types (open-source, closed-source, general-purpose vision-language models, and dedicated reasoning models). The complete protocol and data are provided in the Appendix C.1.3.
>
>
> | Model                  | Consistency Rate(%) | P/N Consistency Rate(%) | Sample Size |
> |------------------------|----------------|--------------------|------------|
> | Gemini2.5 pro          | 87.74          | 87.74              | 106        |
> | Gemini2.5 flash        | 91.00          | 95.00              | 100        |
> | o3                     | 86.54          | 94.23              | 104        |
> | GPT 4.1                | 94.12          | 95.10              | 102        |
> | InternVL3-78B          | 88.24          | 92.16              | 102        |
> | InternVL3-38B          | 92.00          | 96.00              | 100        |
> | Qwen2.5-VL-72B         | 95.05          | 95.05              | 101        |
> | Qwen2.5-VL-32B         | 90.00          | 94.00              | 100        |
> | Phi 4-multimodal       | 89.11          | 98.02              | 101        |
> | MiniCPM-V4             | 95.00          | 97.00              | 100        |
> | MiMo-VL-RL2508         | 93.00          | 96.00              | 100        |
> | GLM 4.5V               | 92.11          | 99.12              | 114        |
> | Kimi-VL-A3B-Thinking   | 90.10          | 95.05              | 101        |
> | **Mean**               | 91.08          | 94.96              | -          |
> | **Std. dev.**          | 2.66           | 2.70               | -          |
> | **95% CI**             | (89.40, 92.75) | (93.26, 96.65)     | -          |
>
> From the table, the consistency rate between LLMs and humans exceeds 90\% with narrow confidence intervals and low standard deviation across different models. This indicates that LLM-as-judge decisions are highly aligned with human judgments, and the variability across evaluated models remains small. Importantly, this alignment remains stable regardless of model family or capability level, suggesting that the reliability of the LLM-as-judge approach is not tied to any particular model architecture. These results demonstrate that the LLM-as-judge paradigm used in FREAK is both reliable and reproducible, and that the performance gap between LLM-based evaluation and human evaluation is minimal.
> ***
> **Minor Weakness**
>
> We thank the reviewer for the careful reading of our paper and the constructive comments. We have addressed all noted minor weaknesses, including:
>
> - Rephrasing the discussion of the saturation of existing benchmarks in Sec. 1.
> - Redesigning the presentation and layout of Table 2 for improved clarity.
> - Correcting typos and formatting issues throughout the paper, and thoroughly checking for any remaining inconsistencies.
>
> We sincerely appreciate the reviewer’s detailed feedback and the time spent evaluating our work.
> ***
> We hope the above responses address your concerns and provide a clearer understanding of our manuscript. We would greatly appreciate it if you could consider increasing your rating.

---

> > ### Comment · Reviewer_EXmZ · 2025-11-25
> >
> > After reading the authors’ responses, my concerns regarding W1, W2, and W3 have been sufficiently addressed. As an additional comment on the authors’ reply, I would appreciate a more detailed analysis of why the ranking of MLLM performance differs when using SeedEdit 3.0 versus Seedream 4.0, as shown in the table in W1. I will maintain my initial rating.

---

> ### Author Response · Authors · 2025-11-26
> **Response to Reviewer EXmZ (3/3)**
>
> Thank you for the acknowledgment, and we are glad that our responses have addressed your concerns. Regarding the additional question about why MLLMs performance differs when using SeedEdit 3.0 versus Seedream 4.0, we conducted further analysis on the generated images and found:
>
> **Even under the same textual modification prompt, different editing models, and sometimes even the same model across trials, may produce images that are semantically aligned but visually implemented in different ways**. Since text prompts typically contain less information than the visual space they condition, the generated images can vary in compositional details despite fulfilling the same modification instruction. These subtle variations alter the difficulty of the corresponding QA, which results in shifts in MLLMs' performance across editing models.
>
> However, despite the inconsistency between editing models, we observe that the accuracy gap for the same MLLM under different editing models is relatively small, This suggests that while editing models introduce visual variance, the overall difficulty level remains similar on average, and the benchmark maintains stable evaluative behavior. **Although the images generated by different editing models are not identical, the stable accuracy across models ensures the robustness and sustainability of FREAK**.
>
> We appreciate your insightful observation and hope that this clarification further strengthens your understanding of the evaluation result.

---

### Official Review · Reviewer_BnmH · 2025-11-01

**Soundness:** 3
**Presentation:** 3
**Contribution:** 2
**Rating:** 4
**Confidence:** 4

**Summary:**

This paper introduces FREAK, a benchmark designed to evaluate fine-grained hallucinations in multimodal large language models (MLLMs). Unlike prior benchmarks focused on object-level errors, FREAK targets subtle, counter-commonsense inconsistencies within photorealistic images across six categories (Detection, Counting, Attribute, Analysis, Position, OCR). Built through an automated generate-then-edit pipeline, FREAK includes human-verified annotations for reliability. Experiments on 17 state-of-the-art models show that even top models achieve only around 45% accuracy, far below the human score of 86.7%, indicating that fine-grained hallucination remains a major challenge. The benchmark also reveals that reasoning (e.g., CoT) can amplify hallucinations rather than mitigate them.

**Strengths:**

* The paper introduces FREAK, a large-scale, AI-generated CCS benchmark that evaluates models across six tasks (Detection, Counting, Attribute, Analysis, Position, OCR), offering a comprehensive assessment of fine-grained hallucination.

* Experiments show that most SOTA models perform far below human baseline on FREAK (≈45% vs. 86.7%), indicating that the benchmark meaningfully exposes robustness gaps under CCS conditions.

* The paper is readable and well-structured, making the motivation, pipeline, and evaluation easy to follow (e.g., the generate-then-edit pipeline and curated Q/A process).

**Weaknesses:**

* Novelty concerns: CCS-style evaluation has appeared in prior benchmarks (e.g., PhD), and the six task types are common in MLLM evaluation, which may narrow the incremental contribution despite FREAK’s fine-grained focus.

* Although FREAK emphasizes photorealistic images with localized CCS edits, real-world usage spans paintings, animations, and other styles; it’s unclear how well the benchmark reflects such domains or whether deduplication alone guarantees sufficient diversity.

* The human baseline would benefit from reporting inter-annotator agreement or additional statistics to aid reproducibility. (The paper notes a 100-participant blind test but gives limited agreement detail.)

* Since an LLM-as-judge paradigm is used for free-form scoring, a discussion/analysis of the judge model’s bias and reliability would strengthen the evaluation methodology.

* Presentation quality: Numerous minor typos/formatting issues (spacing, quotation, punctuation) detract from polish and should be corrected in revision.

**Questions:**

* Why restrict to photorealistic images? Please justify the choice and discuss how conclusions might change for non-photographic domains (e.g., paintings/animations).

* Your analysis suggests CoT often degrades performance/hallucination rates (e.g., Table 4). Would the same trend hold with stronger reasoning strategies (self-refinement, verification-first CoT, or error-detection loops), especially when early visual misperception could be corrected mid-reasoning? [1-3]

**Referneces**:

[1] Gao, Jun, et al. "Interleaved-modal chain-of-thought." Proceedings of the Computer Vision and Pattern Recognition Conference. 2025.

[2] Madaan, Aman, et al. "Self-refine: Iterative refinement with self-feedback." Advances in Neural Information Processing Systems 36 (2023): 46534-46594.

[3] Zhou, Qiji, et al. "Image-of-thought prompting for visual reasoning refinement in multimodal large language models." arXiv preprint arXiv:2405.13872 (2024).

---

> ### Author Response · Authors · 2025-11-20
> **Response to Reviewer BnmH (1/4)**
>
> We thank the reviewer for the detailed and useful review. We hope to address any concerns and questions below.
> ## **Weaknesses**
> ***
> **W1: Novelty Conerns**
> > Novelty concerns: CCS-style evaluation has appeared in prior benchmarks (e.g., PhD), and the six task types are common in MLLM evaluation, which may narrow the incremental contribution despite FREAK’s fine-grained focus.
>
> We appreciate the reviewer’s concern and clarify that our novelty does not lie in merely using CCS-style evaluation, but in addressing the key limitations of prior CCS-style benchmarks.
>
>  - **1. Gap to prior CCS-style benchmarks**
>  Existing CCS-style datasets (e.g., PhD) have limits in image quality, dataset size, and evaluation methodology. Most importantly, the images in these benchmarks do not target fine-grained, photorealistic, visually subtle CCS editing. Our work explicitly fills this gap by constructing a benchmark centered on fine-grained perceptual hallucination, which has not been systematically evaluated in prior work.
>
>  - **2. New generation pipeline enabling fine-grained CCS images**
> We introduce a generate–then–edit pipeline that combines generative and editing models to create high-fidelity, fine-grained CCS images. This capability is absent in prior CCS benchmarks, which rely on artistic illustrations containing evident CCS artifacts. Our pipeline enables scalable, controllable creation of subtle CCS editing that prior work cannot produce.
>
>  - **3. Novel analysis on model reasoning failure modes**
> Beyond dataset construction, we provide the first systematic analysis showing that standard CoT/slow-thinking strategies fail to correct perceptual hallucinations. The reconstructed reasoning traces and statistical behaviors (Sec. 6) offer new insights into MLLM reliability, which is an aspect not studied in prior CCS-style evaluations.
> ***
> **W2: Style and Diversity**
> > Although FREAK emphasizes photorealistic images with localized CCS edits, real-world usage spans paintings, animations, and other styles; it’s unclear how well the benchmark reflects such domains or whether deduplication alone guarantees sufficient diversity.
>
> We thank the reviewer for raising this concern. Our focus on photorealistic images is intentional and grounded in the goal of evaluating fine-grained perceptual hallucinations, which require realistic textures, lighting, and object details. Artistic or stylized images (e.g., paintings, animations) commonly abstract away such cues and therefore cannot reliably expose the subtle visual inconsistencies that our benchmark targets. As stated in Sec. 2, MLLMs demonstrate strong recognition capabilities for these types of artistic illustrations, indicating that such images are insufficient for evaluating hallucination in advanced MLLMs.
>
> Regarding diversity, FREAK achieves substantial coverage within the photorealistic domain. After deduplication, the benchmark contains **1,595 distinct CCS descriptions** across 1,070 unique objects, with minimal overlap except for scenes that differ meaningfully in composition or edit outcomes. This diversity is larger than or comparable to widely used hallucination benchmarks (e.g., HallusionBench (CVPR 2024) ≤346 types, PhD (CVPR 2025) ≤750 types), demonstrating that restricting the style does not limit representativeness.
>
> In summary, although FREAK centers on photorealistic images, this design is essential for probing the fine-grained perceptual failures that current MLLMs still struggle with, while maintaining high diversity and generality within this domain.
> ***
> **W3: Human Baseline Details**
> > The human baseline would benefit from reporting inter-annotator agreement or additional statistics to aid reproducibility. (The paper notes a 100-participant blind test but gives limited agreement detail.)
>
> We have supplemented the experimental results in Appendix C.1.1 of the revised paper to ensure transparency and reproducibility. Specifically, we calculated various metrics of human performance on the multiple-choice questions in FREAK, where "support" indicates the number of items for which a given option is the ground truth. Each question have been answered by one or two different respondents.
>
> | Class | Precision | Recall | F1-Score | Support |
> |-------|-----------|--------|----------|---------|
> | Choice A     | 0.83      | 0.86   | 0.84     | 340     |
> | Choice B     | 0.86      | 0.86   | 0.86     | 335     |
> | Choice C     | 0.87      | 0.83   | 0.85     | 335     |
> | Choice D     | 0      | 0   | 0     | 0       |
> | **Acc** | 0.86         | 0.85     | 0.85     | 1010    |
> | **Macro Avg** | 0.64      | 0.64   | 0.64     | 1010    |
> | **Weighted Avg** | 0.86      | 0.85   | 0.85     | 1010    |
>
> These statistics directly address reproducibility and clarify the consistency of human judgments beyond the average accuracy reported in the main paper. All annotations and scripts for computing agreement will be released after the review process.

---

> ### Author Response · Authors · 2025-11-20
> **Response to Reviewer BnmH (2/4)**
>
> **W4: Reliability of LLM-as-Judge**
> >Since an LLM-as-judge paradigm is used for free-form scoring, a discussion/analysis of the judge model’s bias and reliability would strengthen the evaluation methodology.
>
> To address potential bias and reliability issues in the LLM-as-judge paradigm, we conducted a systematic analysis in **Appendix C.1.3** and now summarize the key findings as below.
>
> We evaluate the reliability by comparing LLM-as-judge decisions with human judgments over 100 randomly sampled items for each of different models (covering open-source, closed-source, non-reasoning MLLMs, and reasoning models). The consistency between human annotators and LLMs exceeds **90%** across all models, and the standard deviations are small, indicating that the judgment outcomes are highly stable and aligned with human preferences. This result demonstrates that the LLM-as-judge setup used in FREAK is both reliable and robust across models, making it an effective method for evaluating free-form responses.
>
> | Model                  | Consistency Rate(%) | P/N Consistency Rate(%) | Sample Size |
> |------------------------|----------------|--------------------|------------|
> | Gemini2.5 pro          | 87.74          | 87.74              | 106        |
> | Gemini2.5 flash        | 91.00          | 95.00              | 100        |
> | o3                     | 86.54          | 94.23              | 104        |
> | GPT 4.1                | 94.12          | 95.10              | 102        |
> | InternVL3-78B          | 88.24          | 92.16              | 102        |
> | InternVL3-38B          | 92.00          | 96.00              | 100        |
> | Qwen2.5-VL-72B         | 95.05          | 95.05              | 101        |
> | Qwen2.5-VL-32B         | 90.00          | 94.00              | 100        |
> | Phi 4-multimodal       | 89.11          | 98.02              | 101        |
> | MiniCPM-V4             | 95.00          | 97.00              | 100        |
> | MiMo-VL-RL2508         | 93.00          | 96.00              | 100        |
> | GLM 4.5V               | 92.11          | 99.12              | 114        |
> | Kimi-VL-A3B-Thinking   | 90.10          | 95.05              | 101        |
> | **Mean**               | 91.08          | 94.96              | -          |
> | **Std. dev.**          | 2.66           | 2.70               | -          |
> | **95% CI**             | (89.40, 92.75) | (93.26, 96.65)     | -          |
> ***
> **W5: Presentation Quality**
> >Presentation quality: Numerous minor typos/formatting issues (spacing, quotation, punctuation) detract from polish and should be corrected in revision.
>
> We sincerely apologize for the typos that appeared in the paper. These errors have been corrected in the updated version, and the content has been carefully proofread. We would like to express our respect for your thorough reading once again!

---

> ### Author Response · Authors · 2025-11-20
> **Response to Reviewer BnmH (3/4)**
>
> ## **Questions**
> ***
> **Q1: Image Style Concerns**
> >Why restrict to photorealistic images? Please justify the choice and discuss how conclusions might change for non-photographic domains (e.g., paintings/animations).
>
> Please see **W2**.
> ***
> **Q2: Stronger Reasoning Stragety**
> >Your analysis suggests CoT often degrades performance/hallucination rates (e.g., Table 4). Would the same trend hold with stronger reasoning strategies (self-refinement, verification-first CoT, or error-detection loops), especially when early visual misperception could be corrected mid-reasoning? [1-3]
>
> Thank you for the insightful question. We have further examined this issue, and our results show that the same trend persists even when employing stronger reasoning strategies beyond standard CoT.
>
> Since the literature lacks a unified taxonomy for advanced reasoning strategies (e.g., verification-first CoT, self-refinement loops, or error-detection chains), we operationalize “stronger reasoning’’ using commonly accepted behavioral signals observed in reasoning-oriented MLLMs: the presence of explicit self-correction triggers (e.g., “wait”, “again”, “let me re-check”), which are widely used by reasoning models to initiate a refinement loop;
> Using these criteria, we analyze the output of Kimi-VL-A3B-thinking and the statistics is shown in the following table. Two consistent findings emerge:
>
> | Kimi-VL-A3B-thinking      | Correct Count | Incorrect Count | Ratio   |
> |---------------------------|---------------|-----------------|---------|
> | Stronger reasoning        | 530           | 909             | 58.31%  |
> | No stronger reasoning     | 1889          | 2672            | 70.70%  |
> | Ratio                     | 28.05%        | 34.02%          |         |
>
> - **Stronger reasoning does not improve accuracy.** \
> The accuracy of responses involving stronger reasoning is 58.31%, noticeably lower than the 70.70% achieved when such strategies are not triggered. This pattern holds across question difficulties, indicating that enhanced reasoning does not mitigate fine-grained hallucination.
>
> - **Stronger reasoning is more frequently invoked in incorrect cases.** \
> A larger proportion of erroneous answers exhibit advanced reasoning behaviors, suggesting that the model attempts to use self-refinement when they exactly detect internal uncertainty. However, despite multiple verification attempts during thinking, it fails to produce text that accurately describes the visual content, exhibiting a phenomenon akin to "hesitation to output accurate information". This phenomenon is discussed in Sec. 6.2 and we exhibit examples in Fig. 22, Fig. 23 and Fig. 24.
>
> Consequently, we conclude that CoT reasoning, even in its more advanced forms, does not yield significant positive effects on fine-grained hallucination tasks.
>
> ***
> We hope the above responses address your concerns and provide a clearer understanding of our manuscript. We would greatly appreciate it if you could consider increasing your rating.

---

> > ### Comment · Reviewer_BnmH · 2025-11-24
> > **Thank you for your response**
> >
> > Thank you for taking the time to provide a detailed response to my comments. I also appreciate you highlighting the relevant material in the appendix — I had initially overlooked some of that content, so the clarification was helpful.
> >
> > That said, I still find it difficult to fully understand the necessity of restricting the focus exclusively to photorealistic images. Your rebuttal does not fully convince me why such a limitation is essential or justified within the broader scope of hallucination analysis.
> >
> > Given this, I will raise my score only moderately to 6.

---

> ### Author Response · Authors · 2025-11-26
> **Response to Reviewer BnmH (4/4)**
>
> Thank you sincerely for your positive feedback. Below we further explain the reasons why we focus on photorealistic images.
> ***
> Our decision is not merely a stylistic preference, but a design choice aligned with the goal of building a benchmark that can still challenge state-of-the-art MLLMs. As shown in Table 1 in paper, existing benchmarks most similar to ours (e.g., Vlind-Bench, PhD-CCS) contain paintings or animation-like CCS images. Current advanced MLLMs already perform strongly on these benchmarks, suggesting that they are becoming less diagnostic for hallucination assessment.
>
> In contrast, our preliminary experiments revealed a clear gap:
>
>  - Humans can easily and consistently detect anomalies in photorealistic CCS images,
>
> - while MLLMs exhibit significantly lower accuracy under the same setting.
>
> We argue that the overall realism of photorealistic images can mislead models, causing them to overlook subtle counter-commonsense details and thus requiring stronger visual attention and factual consistency reasoning. This makes photorealistic CCS images particularly effective for exposing fine-grained hallucination failures.
>
> Therefore, restricting to photorealistic images is not a limitation but a deliberate design to ensure the benchmark remains challenging and relevant for future generations of models. To address diversity concerns, we complemented this stylistic focus with extensive variation in objects and CCS description to maintain wide coverage.
>
> In summary, these non-photorealistic CCS images remain valuable, but photorealistic images are currently more effective at revealing hallucination vulnerabilities in advanced MLLMs, which is why this focus is both necessary and justified.

---

### Author Response · Authors · 2025-12-04
**Rebuttal Summary by Authors**

Dear PCs, ACs, reviewers:

We want to sincerely thank the reviewers for their thoughtful engagement and valuable feedback during rebuttal period, which have further strengthened this work. We are very pleased that all reviewers recommend acceptance (rating >= 6) and acknowledge the significance of our contributions before 11.27.

This paper proposes FREAK, a comprehensive multimodal benchmark designed for fine-grained hallucination assessment in SOTA MLLMs. FREAK addresses key limitations in existing hallucination-evaluation benchmarks and provides the community with a more informative and practically valuable measure of hallucination severity.
***
During the rebuttal period, we responded to all reviewer concerns point-by-point. For general concerns:
- **Reviewer BnmH** and **Reviewer EXmZ** questioned the validity of the LLM-as-judge paradigm; we have conducted additional experiments and reported the results in the Appendix.
- **Reviewer EXmZ**, **Reviewer bnQv**, and **Reviewer Xkgu** raised the possibility of bias introduced by image-editing models; we have performed additional ablation studies to address this.
- **Reviewer BnmH** and **Reviewer bnQv** expressed concerns about the model’s reasoning process; we have expanded the explanation and clarified the mechanism in our rebuttal.
We have also addressed all reviewer-specific questions in detail, which has resolved the corresponding doubts and concerns. The manuscript PDF is updated following reviewers' advice.

**Reviewer BnmH** increased their rating to 6 on Nov. 24, and we provided further clarification to their questions even after the update. After we responded to **Reviewer bnQv** on Nov. 26, bnQv subsequently raised their score to 8 on Nov. 28. **Reviewer Xkgu** did not provide further comments following our responses. These rating updates can be verified directly in each reviewer’s comment history.
***
Overall, we believe we have engaged in constructive and effective communication with the majority of reviewers and have adequately addressed their concerns. We sincerely thank all reviewers for their time and valuable feedback during the rebuttal phase, and we thank the ACs and PCs in advance for their consideration of our work.

---

### Meta-Review · Area_Chair_S9CT · 2026-01-12

**Summary:**

- There are questions about the incremental contribution over existing counter-commonsense benchmarks and whether the focus on fine-grained, photorealistic hallucinations is necessary or reflective of real-world issues.

- Concerns about the LLM-as-Judge evaluation paradigm's bias and reliability. Concerns about potential bias or artifacts introduced by the specific image-editing model affecting benchmark validity.

- Issues with the taxonomy being non-exclusive or non-hierarchical. Why chain-of-thought reasoning harms performance. Adn some questions about dataset bias, scalability, and limitations.

**Reviewer Concerns:**

Addressed Concerns:

- LLM-as-Judge: The authors conducted new experiments comparing LLM judgments with human evaluations across 13 models, showing high consistency rates.

- Image-Editing Model: The authors performed an ablation study using three different editing models. Results showed stable model accuracy and high inter-editor consistency.

- Reasoning: The authors provided an analysis explaining that advanced reasoning often leads to "self-doubt," where models override correct visual perceptions with internal commonsense knowledge.

- Taxonomy: The authors reframed the taxonomy as a "subtask-based design" with refined definitions.



Unaddressed Concerns:

- Justification for Photorealistic-Only Focus (BnmH): Artistic styles are less diagnostic, and reviewer BnmH remained unconvinced.

- Scalability: The dataset size (~1.8K) remains small. The reviewer did not push back further after the explanation.

**Reviewer Scores:**

Reviewer BnmH and Reviewer bnQv may increase the scores, and others may remain the original scores.

---

### Decision · Program_Chairs · 2026-01-26

Accept (Poster)